# THE LIMITING DYNAMICS OF SGD: MODIFIED LOSS, PHASE SPACE OSCILLATIONS, ANOMALOUS DIFFUSION

## ABSTRACT

In this work we explore the limiting dynamics of deep neural networks trained with stochastic gradient descent (SGD). As observed previously, long after performance has converged, networks continue to move through parameter space by a process of anomalous diffusion in which distance travelled grows as a power law in the number of gradient updates with a nontrivial exponent. We reveal an intricate interaction between the hyperparameters of optimization, the structure in the gradient noise, and the Hessian matrix at the end of training that explains this anomalous diffusion. To build this understanding, we first derive a continuous-time model for SGD with finite learning rates and batch sizes as an underdamped Langevin equation. We study this equation in the setting of linear regression, where we can derive exact, analytic expressions for the phase space dynamics of the parameters and their instantaneous velocities from initialization to stationarity. Using the Fokker-Planck equation, we show that the key ingredient driving these dynamics is not the original training loss, but rather the combination of a modified loss, which implicitly regularizes the velocity, and probability currents, which cause oscillations in phase space. We identify qualitative and quantitative predictions of this theory in the dynamics of a ResNet-18 model trained on ImageNet. Through the lens of statistical physics, we uncover a mechanistic origin for the anomalous limiting dynamics of deep neural networks trained with SGD.

## 1 INTRODUCTION

Deep neural networks have demonstrated remarkable generalization across a variety of datasets and tasks. Essential to their success has been a collection of good practices on how to train these models with stochastic gradient descent (SGD). Yet, despite their importance, these practices are mainly based on heuristic arguments and trial and error search. Without a general theory connecting the hyperparameters of optimization, the architecture of the network, and the geometry of the dataset, theory-driven design of deep learning systems is impossible. Existing theoretical works studying this interaction have leveraged the random structure of neural networks at initialization [1, 2, 3] and in their infinite width limits in order to study their dynamics [4, 5, 6, 7, 8]. Here we take a different approach and study the training dynamics of *pre-trained* networks that are ready to be used for inference. By leveraging the mathematical structures found at the end of training, we uncover an intricate interaction between the hyperparameters of optimization, the structure in the gradient noise, and the Hessian matrix that corroborates previously identified empirical behavior such as anomalous limiting dynamics. Not only is understanding the limiting dynamics of SGD a critical stepping stone to building a complete theory for the learning dynamics of neural networks, but recently there have been a series of works demonstrating that the performance of pre-trained networks can be improved through averaging and ensembling [9, 10, 11]. Combining empirical exploration and theoretical tools from statistical physics, we identify and uncover a mechanistic explanation for the limiting dynamics of neural networks trained with SGD.

## 2 DIFFUSIVE BEHAVIOR IN THE LIMITING DYNAMICS OF SGD

A network that has converged in performance will continue to move through parameter space [12, 13, 14, 15]. To demonstrate this behavior, we resume training of pre-trained convolutional networks while tracking the network trajectory through parameter space. Let $\theta_* \in \mathbb{R}^m$ be the parameter vector for a pre-trained network and $\theta_k \in \mathbb{R}^m$ be the parameter vector after $k$ steps of resumed training.

We track two metrics of the training trajectory, namely the local parameter displacement $\delta_k$ between consecutive steps, and the global displacement $\Delta_k$ after $k$ steps from the pre-trained initialization:

$$\delta_k = \theta_k - \theta_{k-1}, \qquad \Delta_k = \theta_k - \theta_*. \qquad (1)$$

As shown in Fig. 1, neither of these differences converge to zero across a variety of architectures, indicating that despite performance convergence, the networks continue to move through parameter space, both locally and globally. The squared norm of the local displacement $\|\delta_k\|_2^2$ remains near a constant value, indicating the network is essentially moving at a constant instantaneous speed. This observation is quite similar to the *"equilibrium"* phenomenon or *"constant angular update"* observed in Li et al. [17] and Wan et al. [13] respectively. However, these works only studied the displacement for parameters immediately preceding a normalization layer. The constant instantaneous speed behavior we observe is for all parameters in the model and is even present in models without normalization layers.

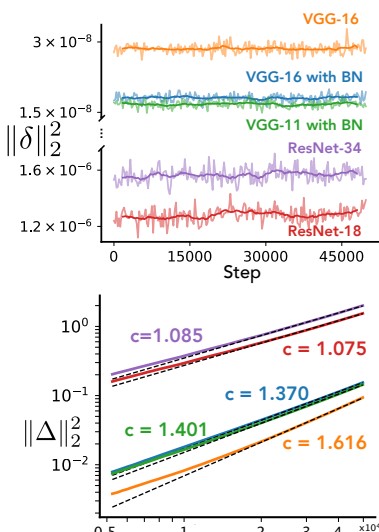

While the squared norm of the local displacement is essentially constant, the squared norm of the global displacement $\|\Delta_k\|_2^2$ is monotonically growing for all networks, implying even once trained, the network continues to diverge from where it has been. Indeed Fig. 1 indicates a power law relationship between global displacement and number of steps, given by $\|\Delta_k\|_2^2 \propto k^c$. As we'll see in section 8, this relationship is indicative of anomalous diffusion where $c$ corresponds to the anomalous diffusion exponent. Standard Brownian motion corresponds to $c = 1$. Similar observation were made by Baity-Jesi et al. [14] who noticed distinct phases of the training trajectory evident in the dynamics of the global displacement and Chen et al. [15] who found that the exponent of diffusion changes through the course of training. A parallel observation is given by Hoffer et al. [18] for the beginning of training, where they measure the global dis-

Figure 1: **Despite performance convergence, the network continues to move through parameter space.** We plot the squared Euclidean norm for the local and global displacement ($\delta_k$ and $\Delta_k$) of five classic convolutional neural network architectures. The networks are standard Pytorch models pre-trained on ImageNet [16]. Their training is resumed for 10 additional epochs. We show the global displacement on a log-log plot where the slope of the least squares line $c$ is the exponent of the power law $\|\Delta_k\|_2^2 \propto k^c$. See appendix H for experimental details.

placement from the initialization of an untrained network and observe a rate $\propto \log(k)$, a form of ultra-slow diffusion. These empirical observations raise the natural questions, *where is the network moving to and why?* To answer these questions we will build a diffusion based theory of SGD, study these dynamics in the setting of linear regression, and use lessons learned in this fundamental setting to understand the limiting dynamics of neural networks.

## 3 RELATED WORK

There is a long line of literature studying both theoretically and empirically the learning dynamics of deep neural networks trained with SGD. Our analysis and experiments build upon this literature.

**Continuous models for SGD.** Many works consider how to improve the classic gradient flow model for SGD to more realistically reflect momentum [19], discretization due to finite learning rates [20, 21], and stochasticity due to random batches [22, 23]. One line of work has studied the dynamics of networks in their infinite width limits through dynamical mean field theory [24, 25, 26, 27], while a different approach has used stochastic differential equations (SDEs) to model SGD directly, the approach we take in this work. However, recently, the validity of this approach has been questioned. The main argument, as nicely explained in Yaida [28], is that most SDE approximations simultaneously assume that $\Delta t \to 0^+$, while maintaining that the learning rate $\eta = \Delta t$ is finite. The works Simsekli et al. [29] and Li et al. [30] have questioned the correctness of the using the central limit theorem (CLT) to model the gradient noise as Gaussian, arguing respectively that the heavy-tailed structure in the gradient noise and the weak dependence between batches leads the CLT to break down. In our work, we maintain the CLT assumption holds, which we discuss fur-

ther in appendix A, but importantly we avoid the pitfalls of many previous SDE approximations by simultaneously modeling the effect of finite learning rates and stochasticity.

**Limiting dynamics.** A series of works have applied SDE models of SGD to study the limiting dynamics of neural networks. In the seminal work by Mandt et al. [31], the limiting dynamics were modeled with a multivariate Ornstein-Uhlenbeck process by combining a first-order SDE model for SGD with assumptions on the geometry of the loss and covariance matrix for the gradient noise. This analysis was extended by Jastrzębski et al. [12] through additional assumptions on the covariance matrix to gain tractable insights and applied by Ali et al. [32] to the simpler setting of linear regression, which has a quadratic loss. A different approach was taken by Chaudhari and Soatto [33], which did not formulate the dynamics as an OU process, nor assume directly a structure on the loss or gradient noise. Rather, this analysis studied the same first-order SDE via the Fokker-Planck equation to propose the existence of a modified loss and probability currents driving the limiting dynamics, but did not provide explicit expressions. Our analysis deepens and combines ideas from all these works, where our key insight is to lift the dynamics into phase space. By studying the dynamics of the parameters and their velocities, and by applying the analysis first in the setting of linear regression where assumptions are provably true, we are able to identify analytic expressions and explicit insights which lead to concrete predictions and testable hypothesis.

**Stationary dynamics.** A different line of work avoids modeling the limiting dynamics of SGD with an SDE and instead chooses to leverage the property of stationarity. These works [28, 34, 35, 36] assume that eventually the probability distribution governing the model parameters reaches stationarity such that the discrete SGD process is simply sampling from this distribution. Yaida [28] used this approach to derive fluctuation-dissipation relations that link measurable quantities of the parameters and hyperparameters of SGD. Liu et al. [35] used this approach to derive properties for the stationary distribution of SGD with a quadratic loss. Similar to our analysis, this work identifies that the stationary distribution for the parameters reflects a modified loss function dependent on the relationship between the covariance matrix of the gradient noise and the Hessian matrix for the original loss.

**Empirical exploration.** Another set of works analyzing the limiting dynamics of SGD has taken a purely empirical approach. Building on the intuition that flat minima generalize better than sharp minima, Keskar et al. [37] demonstrated empirically that the hyperparameters of optimization influence the eigenvalue spectrum of the Hessian matrix at the end of training. Many subsequent works have studied the Hessian eigenspectrum during and at the end of training. Jastrzębski et al. [38], Cohen et al. [39] studied the dynamics of the top eigenvalues during training. Sagun et al. [40], Papyan [41], Ghorbani et al. [42] demonstrated the spectrum has a bulk of values near zero plus a small number of larger outliers. Gur-Ari et al. [43] demonstrated that the learning dynamics are constrained to the subspace spanned by the top eigenvectors, but found no special properties of the dynamics within this subspace. In our work we also determine that the top eigensubspace of the Hessian plays a crucial role in the limiting dynamics and by projecting the dynamics into this subspace in phase space, we see that the motion is not random, but consists of incoherent oscillations leading to anomalous diffusion.

## 4 MODELING SGD AS AN UNDERDAMPED LANGEVIN EQUATION

Following the route of previous works [31, 12, 33] studying the limiting dynamics of neural networks, we first seek to model SGD as a continuous stochastic process. We consider a network parameterized by $\theta \in \mathbb{R}^m$, a training dataset $\{x_1, \ldots, x_N\}$ of size $N$, and a training loss $\mathcal{L}(\theta) = \frac{1}{N} \sum_{i=1}^{N} \ell(\theta, x_i)$ with corresponding gradient $g(\theta) = \frac{\partial \mathcal{L}}{\partial \theta}$. The state of the network at the $k^{\text{th}}$ step of training is defined by the position vector $\theta_k$ and velocity vector $v_k$ of the same dimension. The gradient descent update with learning rate $\eta$, momentum $\beta$, and weight decay $\lambda$ is given by

$$v_{k+1} = \beta v_k - g(\theta_k) - \lambda \theta_k, \qquad \theta_{k+1} = \theta_k + \eta v_{k+1}, \qquad (2)$$

where we initialize the network such that $v_0 = 0$ and $\theta_0$ is the parameter initialization. In order to understand the dynamics of the network through position and velocity space, which we will refer to as *phase space*, we express these discrete recursive equations as the discretization of some unknown ordinary differential equation (ODE), sometimes referred to as a modified equation as in [44, 20]. While this ODE models the gradient descent process even at finite learning rates, it fails to account for the stochasticity introduced by choosing a random batch $\mathcal{B}$ of size $S$ drawn uniformly from the set of $N$ training points. This sampling yields the stochastic gradient $g_{\mathcal{B}}(\theta) = \frac{1}{S} \sum_{i \in \mathcal{B}} \nabla \ell(\theta, x_i)$. To model this effect, we make the following assumption:

**Assumption 1** (CLT). *We assume the batch gradient is a noisy version of the true gradient such that $g_{\mathcal{B}}(\theta) - g(\theta)$ is a Gaussian random variable with mean $0$ and covariance $\frac{1}{S}\Sigma(\theta)$.*

Incorporating this model of stochastic gradients into the previous finite difference equation and applying the stochastic counterparts to Euler discretizations, results in the standard drift-diffusion stochastic differential equation (SDE), referred to as an *underdamped Langevin equation*,

$$d \begin{bmatrix} \theta \\ v \end{bmatrix} = \begin{bmatrix} v \\ -\frac{2}{\eta(1+\beta)}\left(g(\theta) + \lambda\theta + (1-\beta)v\right) \end{bmatrix} dt + \begin{bmatrix} 0 & 0 \\ 0 & \frac{2}{\sqrt{\eta S}(1+\beta)}\sqrt{\Sigma(\theta)} \end{bmatrix} dW_t, \quad (3)$$

where $W_t$ is a standard Wiener process. This is the continuous model we will study in this work:

**Assumption 2** (SDE). *We assume the underdamped Langevin equation (3) accurately models the trajectory of the network driven by SGD through phase space such that $\theta(\eta k) \approx \theta_k$ and $v(\eta k) \approx v_k$.*

See appendix A for further discussion on the nuances of modeling SGD with an SDE.

## 5 LINEAR REGRESSION WITH SGD IS AN ORNSTEIN-UHLENBECK PROCESS

Equipped with a model for SGD, we seek to understand its dynamics in the fundamental setting of linear regression, one of the few cases where we have a complete model for the interaction of the dataset, architecture, and optimizer. Let $X \in \mathbb{R}^{N \times d}$ be the input data, $Y \in \mathbb{R}^N$ be the output labels, and $\theta \in \mathbb{R}^d$ be our vector of regression coefficients. The least squares loss is the convex quadratic loss $\mathcal{L}(\theta) = \frac{1}{2N}\|Y - X\theta\|^2$ with gradient $g(\theta) = H\theta - b$, where $H = \frac{X^\intercal X}{N}$ and $b = \frac{X^\intercal Y}{N}$. Plugging this expression for the gradient into the underdamped Langevin equation (3), and rearranging terms, results in the multivariate Ornstein-Uhlenbeck (OU) process,

$$d \begin{bmatrix} \theta_t \\ v_t \end{bmatrix} = -\underbrace{\begin{bmatrix} 0 & -I \\ \frac{2}{\eta(1+\beta)}(H + \lambda I) & \frac{2(1-\beta)}{\eta(1+\beta)}I \end{bmatrix}}_{A} \left( \begin{bmatrix} \theta_t \\ v_t \end{bmatrix} - \begin{bmatrix} \mu \\ 0 \end{bmatrix} \right) dt + \sqrt{2\kappa^{-1}} \sqrt{\underbrace{\begin{bmatrix} 0 & 0 \\ 0 & \frac{2(1-\beta)}{\eta(1+\beta)}\Sigma(\theta) \end{bmatrix}}_{D}} dW_t,$$

$$(4)$$

where $A$ and $D$ are the drift and diffusion matrices respectively, $\kappa = S(1 - \beta^2)$ is an inverse temperature constant, and $\mu = (H + \lambda I)^{-1}b$ is the ridge regression solution. The solution to an OU process is a Gaussian process. By solving for the temporal dynamics of the first and second moments of the process, we can obtain an analytic expression for the trajectory at any time $t$. In particular, we can decompose the trajectory as the sum of a deterministic and stochastic component defined by the first and second moments respectively.

**Deterministic component.** Using the form of $A$ we can decompose the expectation as a sum of harmonic oscillators in the eigenbasis $\{q_1, \ldots, q_m\}$ of the Hessian,

$$\mathbb{E}\left[\begin{bmatrix} \theta_t \\ v_t \end{bmatrix}\right] = \begin{bmatrix} \mu \\ 0 \end{bmatrix} + \sum_{i=1}^{m}\left(a_i(t)\begin{bmatrix} q_i \\ 0 \end{bmatrix} + b_i(t)\begin{bmatrix} 0 \\ q_i \end{bmatrix}\right). \quad (5)$$

Here the coefficients $a_i(t)$ and $b_i(t)$ depend on the optimization hyperparameters $\eta, \beta, \lambda, S$ and the respective eigenvalue of the Hessian $\rho_i$ as further explained in appendix F. We verify this expression nearly perfectly matches empirics on complex datasets under various hyperparameter settings as shown in Fig. 2.

**Stochastic component.** The cross-covariance of the process between two points in time $t \leq s$, is

$$\text{Cov}\left(\begin{bmatrix} \theta_t \\ v_t \end{bmatrix}, \begin{bmatrix} \theta_s \\ v_s \end{bmatrix}\right) = \kappa^{-1}\left(B - e^{-At}Be^{-A^\intercal t}\right)e^{A^\intercal(t-s)}, \quad (6)$$

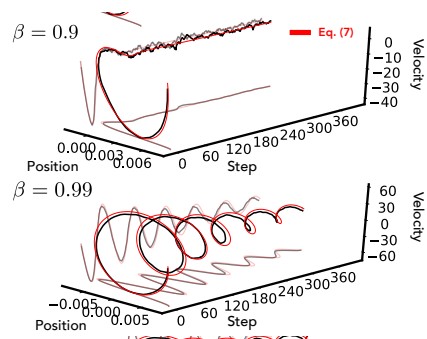

Figure 2: **Oscillatory dynamics in linear regression.** We train a linear network to perform regression on the CIFAR-10 dataset by using an MSE loss on the one-hot encoding of the labels. We compute the hessian of the loss, as well as its top eigenvectors. The position and velocity trajectories are projected onto the first eigenvector of the hessian and visualized in black. The theoretically derived mean, equation (5), is shown in red. The top and bottom panels demonstrate the effect of varying momentum on the oscillation mode.

where $B$ solves the Lyapunov equation $AB + BA^\intercal = 2D$. In order to gain analytic expressions for $B$ in terms of the optimization hyperparameters, eigendecomposition of the Hessian, and covariance of the gradient noise, we must introduce the following assumption:

**Assumption 3** (Simultaneously Diagonalizable). *We assume the covariance of the gradient noise is spatially independent $\Sigma(\theta) = \Sigma$ and commutes with the Hessian $H\Sigma = \Sigma H$, therefore sharing a common eigenbasis.*

## 6 Understanding Stationarity via the Fokker-Planck Equation

The OU process is unique in that it is one of the few SDEs which we can solve exactly. As shown in section 5, we were able to derive exact expressions for the dynamics of linear regression trained with SGD from initialization to stationarity by simply solving for the first and second moments. While the expression for the first moment provides an understanding of the intricate oscillatory relationship in the deterministic component of the process, the second moment, driving the stochastic component, is much more opaque. An alternative route to solving the OU process that potentially provides more insight is the Fokker-Planck equation.

The Fokker-Planck (FP) equation is a PDE describing the time evolution for the probability distribution of a particle governed by Langevin dynamics. For an arbitrary potential $\Phi$ and diffusion matrix $D$, the Fokker-Planck equation (under an Itô integration prescription) is

$$\partial_t p = \nabla \cdot \underbrace{\left( \nabla \Phi p + \nabla \cdot \left( \kappa^{-1} D p \right) \right)}_{-J}, \tag{7}$$

where $p$ represents the time-dependent probability distribution, and $J$ is a vector field commonly referred to as the probability current. The FP equation is especially useful for explicitly solving for the stationary solution, assuming one exists, of the Langevin dynamics. The stationary solution $p_{ss}$ by definition obeys $\partial_t p_{ss} = 0$ or equivalently $\nabla \cdot J_{ss} = 0$. From this second definition we see that there are two distinct settings of stationarity: *detailed balance* when $J_{ss} = 0$ everywhere, or *broken detailed balance* when $\nabla \cdot J_{ss} = 0$ and $J_{ss} \neq 0$.

For a general OU process, the potential is a convex quadratic function $\Phi(x) = x^\intercal A x$ defined by the drift matrix $A$. When the diffusion matrix is isotropic ($D \propto I$) and spatially independent ($\nabla \cdot D = 0$) the resulting stationary solution is a *Gibbs distribution* $p_{ss}(x) \propto e^{-\kappa \Phi(x)}$ determined by the original loss $\Phi(x)$ and is in detailed balance. Lesser known properties of the OU process arise when the diffusion matrix is anisotropic or spatially dependent [45, 46]. In this setting the solution is still a Gaussian process, but the stationary solution, if it exists, is no longer defined by the Gibbs distribution of the original loss $\Phi(x)$, but actually a modified loss $\Psi(x)$. Furthermore, the stationary solution may be in broken detailed balance leading to a non-zero probability current $J_{ss}(x)$. Depending on the relationship between the drift matrix $A$ and the diffusion matrix $D$ the resulting dynamics of the OU process can have very nontrivial behavior.

In the setting of linear regression, anisotropy in the data distribution will lead to anisotropy in the gradient noise and thus an anisotropic diffusion matrix. This implies that for most datasets we should expect that the SGD trajectory is not driven by the original least squares loss, but by a modified loss and converges to a stationary solution with broken detailed balance, as predicted by Chaudhari and Soatto [33]. Using the explicit expressions for the drift $A$ and diffusion $D$ matrices we can compute analytically the modified loss and stationary probability current,

$$\Psi(\theta, v) = \left( \begin{bmatrix} \theta \\ v \end{bmatrix} - \begin{bmatrix} \mu \\ 0 \end{bmatrix} \right)^\intercal \left( \frac{U}{2} \right) \left( \begin{bmatrix} \theta \\ v \end{bmatrix} - \begin{bmatrix} \mu \\ 0 \end{bmatrix} \right), \quad J_{ss}(\theta, v) = -QU \left( \begin{bmatrix} \theta \\ v \end{bmatrix} - \begin{bmatrix} \mu \\ 0 \end{bmatrix} \right) p_{ss}, \tag{8}$$

where $Q$ is a skew-symmetric matrix and $U$ is a positive definite matrix defined as,

$$Q = \begin{bmatrix} 0 & -\Sigma(\theta) \\ \Sigma(\theta) & 0 \end{bmatrix}, \qquad U = \begin{bmatrix} \frac{2}{\eta(1+\beta)} \Sigma(\theta)^{-1} \left( H + \lambda I \right) & 0 \\ 0 & \Sigma(\theta)^{-1} \end{bmatrix}. \tag{9}$$

These new fundamental matrices, $Q$ and $U$, relate to the original drift $A$ and diffusion $D$ matrices through the unique decomposition $A = (D + Q)U$, introduced by Ao [47] and Kwon et al. [48]. Using this decomposition we can easily show that $B = U^{-1}$ solves the Lyapunov equation and indeed the stationary solution $p_{ss}$ is the Gibbs distribution defined by the modified loss $\Psi(\theta, v)$ in equation (8). Further, the stationary cross-covariance solved in section 5 reflects the oscillatory dynamics introduced by the stationary probability currents $J_{ss}(\theta, v)$ in equation (8). Taken together, we gain the intuition that the limiting dynamics of SGD in linear regression are driven by a modified loss subject to oscillatory probability currents.

## 7 EVIDENCE OF A MODIFIED LOSS AND OSCILLATIONS IN DEEP LEARNING

Does the theory derived in the linear regression setting (sections 5, 6) help explain the empirical phenomena observed in the non-linear setting of deep neural networks (section 2)? In order for the theory built in the previous sections to apply to the limiting dynamics of neural networks, we must introduce simplifying assumptions on the loss landscape and gradient noise at the end of training:

**Assumption 4** (Quadratic Loss). *We assume that at the end of training the loss for a neural network can be approximated by the quadratic loss* $\mathcal{L}(\theta) = (\theta - \mu)^{\mathsf{T}} \left( \frac{H}{2} \right) (\theta - \mu)$, *where* $H \succeq 0$ *is the training loss Hessian and* $\mu$ *is some unknown mean vector, corresponding to a local minimum.*

**Assumption 5** (Covariance Structure). *We assume the covariance of the gradient noise is proportional to the Hessian of the quadratic loss* $\Sigma(\theta) = \sigma^2 H$ *where* $\sigma \in \mathbb{R}^+$ *is some unknown scalar.*

Under these simplifications, then the expressions derived in the linear regression setting would apply to the limiting dynamics of deep neural networks and depend only on quantities that we can easily estimate empirically. Of course, these simplifications are quite strong, but without arguing their theoretical validity, we can empirically test their qualitative implications: (1) a modified isotropic loss driving the limiting dynamics through parameter space, (2) implicit regularization of the velocity trajectory, and (3) oscillatory phase space dynamics determined by the Hessian eigen-structure.

**Modified loss.** As discussed in section 6, due to the anisotropy of the diffusion matrix, the loss landscape driving the dynamics at the end of training is not the original training loss $\mathcal{L}(\theta)$, but a modified loss $\Psi(\theta, v)$ in phase space. As shown in equation (8), the modified loss decouples into a term $\Psi_\theta$ that only depends on the parameters $\theta$ and a term $\Psi_v$ that only depends on the velocities $v$. Under assumption 5, the parameter dependent component is proportional to the convex quadratic,

$$\Psi_\theta \propto (\theta - \mu)^{\mathsf{T}} \left( \frac{H^{-1}(H + \lambda I)}{\eta(1 + \beta)} \right) (\theta - \mu). \tag{10}$$

This quadratic function has the same mean $\mu$ as the training loss, but a different curvature. Using this expression, notice that when $\lambda \approx 0$, the modified loss is *isotropic* in the column space of $H$, regardless of what the nonzero eigenspectrum of $H$ is. This striking prediction suggests that no matter how anisotropic the original training loss – as reflected by poor conditioning of the Hessian eigenspectrum – the training trajectory of the network will behave isotropically, since it is driven not by the original anisotropic loss, but a modified *isotropic* loss.

We test this prediction by studying the limiting dynamics of a pre-trained ResNet-18 model with batch normalization that we continue to train on ImageNet according to the last setting of its hyper-parameters [49]. Let $\theta_*$ represent the initial pre-trained parameters of the network, depicted with the white dot in figures 3 and 4.

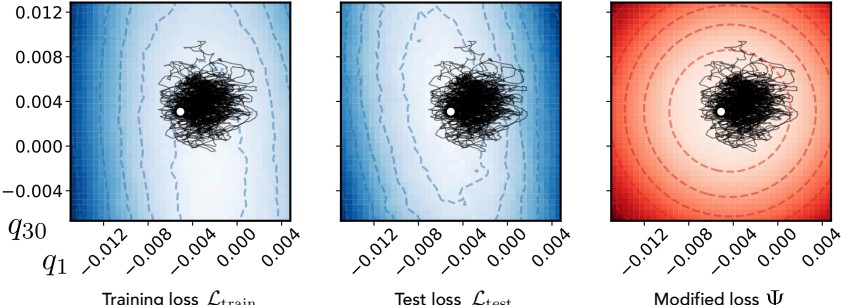

Figure 3: **The training trajectory behaves isotropically, regardless of the training loss.** We resume training of a pre-trained ResNet-18 model on ImageNet and project its parameter trajectory (black line) onto the space spanned by the eigenvectors of its pre-trained Hessian $q_1, q_{30}$ (with eigenvalue ratio $\rho_1/\rho_{30} \simeq 6$). We sample the training and test loss within the same 2D subspace and visualize them as a heatmap in the left and center panels respectively. We visualize the modified loss computed from the eigenvalues $(\rho_1, \rho_{30})$ and optimization hyperparameters according to equation (10) in the right plot. Note the projected trajectory is isotropic, despite the anisotropy of the training and test loss.

We estimate[1] the top thirty eigenvectors $q_1, \ldots, q_{30}$ of the Hessian matrix $H_*$ evaluated at $\theta_*$ and project the limiting trajectory for the parameters onto the plane spanned by the top $q_1$ and bottom $q_{30}$ eigenvectors to maximize the illustrated anisotropy with our estimates. We sample the train and test loss in this subspace for a region around the projected trajectory. Additionally, using the hyperparameters of the optimization, the eigenvalues $\rho_1$ and $\rho_{30}$, and the estimate for the mean $\mu = \theta_* - H_*^{-1} g_*$ ($g_*$ is the gradient evaluated at $\theta_*$), we also sample from the modified loss equation (10) in the same region. Figure 3 shows the projected parameter trajectory on the sampled train, test and modified losses. Contour lines of both the train and test loss exhibit anisotropic structure, with sharper curvature along eigenvector $q_1$ compared to eigenvector $q_{30}$, as expected. However, as predicted, the trajectory appears to cover both directions equally. This striking isotropy of the trajectory within a highly anisotropic slice of the loss landscape indicates qualitatively that the trajectory evolves in a modified isotropic loss landscape.

**Implicit velocity regularization.** A second qualitative prediction of the theory is that the velocity is regulated by the inverse Hessian of the training loss. Of course there are no explicit terms in either the train or test losses that depend on the velocity. Yet, the modified loss contains a component, $\Psi_v \propto v^\intercal H^{-1} v$, that only depends on the velocities This additional term can be understood as a form of implicit regularization on the velocity trajectory. Indeed, when we project the velocity trajectory onto the plane spanned by the $q_1$ and $q_{30}$ eigenvectors, as shown in Fig. 4, we see that the trajectory closely resembles the curvature of the inverse Hessian $H^{-1}$. The modified loss is effectively penalizing SGD for moving in eigenvectors of the Hessian with small eigenvalues. A similar qualitative effect was recently proposed by Barrett and Dherin [21] as a consequence of the discretization error due to finite learning rates.

**Phase space oscillations.** A final implication of the theory is that at stationarity the network is in broken detailed balance leading to non-zero probability currents flowing through phase space:

$$J_{ss}(\theta, v) = \begin{bmatrix} v \\ -\frac{2}{\eta(1+\beta)} \left( H + \lambda I \right) \left( \theta - \mu \right) \end{bmatrix} p_{ss}. \quad (11)$$

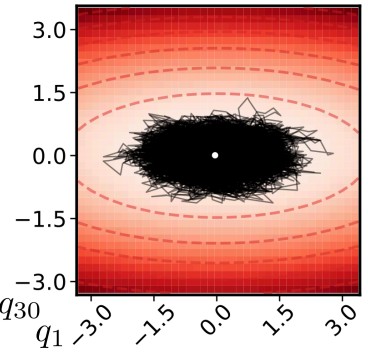

Figure 4: **Implicit velocity regularization defined by the inverse Hessian.** The shape of the projected velocity trajectory closely resembles the contours of the modified loss $\Psi_v$.

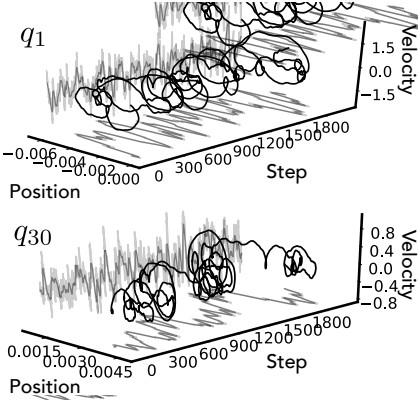

Figure 5: **Phase space oscillations are determined by the eigendecomposition of the Hessian.** We visualize the projected position and velocity trajectories in phase space. The top and bottom panels show the projections onto $q_1$ and $q_{30}$ respectively. Oscillations at different rates are distinguishable for the different eigenvectors and were verified by comparing the dominant frequencies in the fast Fourier transform of the trajectories.

These probability currents encourage oscillatory dynamics in the phase space planes characterized by the eigenvectors of the Hessian, at rates proportional to their eigenvalues. We consider the same projected trajectory of the ResNet-18 model visualized in figures 3 and 4, but plot the trajectory in phase space for the two eigenvectors $q_1$ and $q_{30}$ separately. Shown in Fig. 5, we see that both trajectories look like noisy clockwise rotations. Qualitatively, the trajectories for the different eigenvectors appear to be rotating at different rates.

The integral curves of the stationary probability current are one-dimensional paths confined to level sets of the modified loss. These paths might cross themselves, in which case they are limit cycles, or they could cover the entire surface of the level sets, in which case they are space-filling curves. This distinction depends on the relative frequencies of the oscillations, as determined by the pairwise

---

[1]To estimate the eigenvectors of $H_*$ we use subspace iteration, and limit ourselves to 30 eigenvectors to constrain computation time. See appendix H for details.

ratios of the eigenvalues of the Hessian. For real-world datasets, with a large spectrum of incommensurate frequencies, we expect to be in the latter setting, thus contradicting the suggestion that SGD in deep networks converges to limit cycles, as claimed in Chaudhari and Soatto [33].

## 8 UNDERSTANDING THE DIFFUSIVE BEHAVIOUR OF THE LIMITING DYNAMICS

Taken together the empirical results shown in section 7 indicate that many of the same qualitative behaviors of SGD identified theoretically for linear regression are evident in the limiting dynamics of neural networks. Can this theory quantitatively explain the results we identified in section 2?

**Constant instantaneous speed.** As noted in section 2, we observed that at the end of training, across various architectures, the squared norm of the local displacement $\|\delta_t\|_2^2$ remains essentially constant. Assuming the limiting dynamics are described by the stationary solution the expectation of the local displacement is

$$\mathbb{E}_{ss}\left[\|\delta_t\|^2\right] = \frac{\eta^2}{S(1-\beta^2)}\sigma^2 \mathrm{tr}\left(H\right), \tag{12}$$

as derived in appendix G. We cannot test this prediction directly as we do not know $\sigma^2$ and computing $\mathrm{tr}(H)$ is computationally prohibitive. However, we can estimate $\sigma^2\mathrm{tr}(H)$ by resuming training for a model, measuring the average $\|\delta_t\|^2$, and then inverting equation (12). Using this *single* estimate, we find that for a sweep of models with varying hyperparameters, equation (12) accurately predicts their instantaneous speed. Indeed, Fig. 6 shows an exact match between the empirics and theory, which strongly suggests that despite changing hyperparameters at the end of training, the model remains in the same quadratic basin.

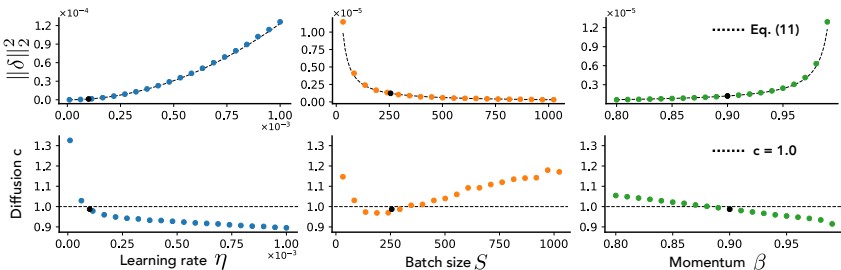

Figure 6: **Understanding how the hyperparameters of optimization influence the diffusion.** We resume training of pre-trained ResNet-18 models on ImageNet using a range of learning rates, batch sizes, and momentum coefficients, tracking $\|\delta_t\|^2$ and $\|\Delta_t\|^2$. Starting from the default hyperparameters, namely $\eta = 1e - 4$, $S = 256$, and $\beta = 0.9$, we vary each one while keeping the others fixed. The top row shows the measured $\|\delta_t\|^2$ in color, with the default hyperparameter setting highlighted in black. The dotted line depicts the predicted value from equation (12). The bottom row shows the estimated exponent $c$ found by fitting a line to the $\|\Delta_t\|^2$ trajectories on a log-log plot. The dotted line shows $c = 1$, corresponding to standard diffusion.

**Exponent of anomalous diffusion.** The expected value for the global displacement under the stationary solution can also be analytically expressed in terms of the optimization hyperparameters and the eigendecomposition of the Hessian as,

$$\mathbb{E}_{ss}\left[\|\Delta_t\|^2\right] = \frac{\eta^2}{S(1-\beta^2)}\sigma^2\left(\mathrm{tr}\left(H\right)t + 2t\sum_{k=1}^{t}\left(1-\frac{k}{t}\right)\sum_{l=1}^{m}\rho_l\mathrm{C}_l(k)\right), \tag{13}$$

where $\mathrm{C}_l(k)$ is a trigonometric function describing the velocity of a harmonic oscillator with damping ratio $\zeta_l = (1-\beta)/\sqrt{2\eta(1+\beta)(p_l+\lambda)}$, see appendix G for details. As shown empirically in section 2, the squared norm $\|\Delta_t\|^2$ monotonically increases as a power law in the number of steps, suggesting its expectation is proportional to $t^c$ for some unknown, constant $c$. The exponent $c$ determines the regime of diffusion for the process. When $c = 1$, the process corresponds to standard Brownian diffusion. For $c > 1$ or $c < 1$ the process corresponds to anomalous super-diffusion or sub-diffusion respectively. Unfortunately, it is not immediately clear how to extract the explicit

exponent $c$ from equation (13). However, by exploring the functional form of $C_l(k)$ and its relationship to the hyperparameters of optimization through the damping ratio $\zeta_l$, we can determine overall trends in the diffusion exponent $c$.

Akin to how the exponent $c$ determines the regime of diffusion, the damping ratio $\zeta_l$ determines the regime for the harmonic oscillator describing the stationary velocity-velocity correlation in the $l^{\text{th}}$ eigenvector of the Hessian. When $\zeta_l = 1$, the oscillator is critically damped implying the velocity correlations converge to zero as quickly as possible. In the extreme setting of $C_l(k) = 0$ for all $l, k$, then equation (13) simplifies to standard Brownian diffusion, $\mathbb{E}_{ss}\left[\|\Delta_t\|^2\right] \propto t$. When $\zeta_l > 1$, the oscillator is overdamped implying the velocity correlations dampen slowly and remain positive even over long temporal lags. Such long lasting temporal correlations in velocity lead to faster global displacement. Indeed, in the extreme setting of $C_l(k) = 1$ for all $l, k$, then equation (13) simplifies to a form of anomalous super-diffusion, $\mathbb{E}_{ss}\left[\|\Delta_t\|^2\right] \propto t^2$. When $\zeta_l < 1$, the oscillator is underdamped implying the velocity correlations will oscillate quickly between positive and negative values. Indeed, the only way equation (13) could describe anomalous sub-diffusion is if $C_l(k)$ took on negative values for certain $l, k$.

Using the same sweep of models described previously, we can empirically confirm that the optimization hyperparameters each influence the diffusion exponent $c$. As shown in Fig. 6, the learning rate, batch size, and momentum can each independently drive the exponent $c$ into different regimes of anomalous diffusion. Notice how the influence of the learning rate and momentum on the diffusion exponent $c$ closely resembles their respective influences on the damping ratio $\zeta_l$. Interestingly, a *larger* learning rate leads to underdamped oscillations, and the resultant temporal velocities' anti-correlations *reduce* the exponent of anomalous diffusion. Thus contrary to intuition, a *larger* learning rate actually leads to *slower* global transport in parameter space. The batch size on the other hand, has no influence on the damping ratio, but leads to an interesting, non-monotonic influence on the diffusion exponent. Overall, the hyperparameters of optimization and eigenspectrum of the Hessian all conspire to govern the degree of anomalous diffusion at the end of training.

## 9 DISCUSSION

Through combined empirics and theory based on statistical physics, we uncovered an intricate interplay between the optimization hyperparameters, structure in the gradient noise, and the Hessian matrix at the end of training.

**Significance.** The significance of our work lies in (1) the identification/verification of multiple empirical phenomena (constant instantaneous speed, anomalous diffusion in global displacement, isotropic parameter exploration despite anisotopic loss, velocity regularization, and *slower* global parameter exploration with *faster* learning rates) present in the limiting dynamics of deep neural networks, (2) the emphasis on studying the dynamics in velocity space in addition to parameter space, and (3) concrete quantitative as well as qualitative predictions of an SDE based theory that we empirically verified in deep networks trained on large scale datasets (indeed some of the above nontrivial phenomena were *predictions* of this theory). Of course, these contributions directly build upon a series of related works studying the immensely complex process of deep learning. To this end, we further clarify the originality of our contributions with respect to some relevant works.

**Originality.** The empirical phenomena we present provide novel insight with respect to the works of Wan et al. [13], Hoffer et al. [18], and Chen et al. [15]. We observe that all parameters in the network (not just those with scale symmetry) move at a constant instantaneous speed at the end of training and diffuse anomalously at rates determined by the hyperparameters of optimization. In contrast to the work by Liu et al. [35], we modeled the entire SGD process as an OU process which allows us to provide insight into the transient dynamics and identify oscillations in parameter and velocity space. We build on the theoretical framework used by Chaudhari and Soatto [33] and provide explicit expressions for the limiting dynamics in the simplified linear regression setting and conclude that the oscillations present in the limiting dynamics are more likely to be space-filling curves (and not limit cycles) in deep learning due to many incommensurate oscillations.

Overall, by identifying key phenomena, explaining them in a simpler setting, deriving predictions of new phenomena, and providing evidence for these predictions at scale, we are furthering the scientific study of deep learning. We hope our newly derived understanding of the limiting dynamics of SGD, and its dependence on various important hyperparameters like batch size, learning rate, and momentum, can serve as a basis for future work that can turn these insights into algorithmic gains.

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

# A  MODELING SGD WITH AN SDE

As explained in section 4, in order to understand the dynamics of stochastic gradient descent we build a continuous Langevin equation in phase space modeling the effect of discrete updates and stochastic batches simultaneously.

## A.1  MODELING DISCRETIZATION

To model the discretization effect we assume that the system of update equations (2) is actually a discretization of some unknown ordinary differential equation. To uncover this ODE, we combine the two update equations in (2), by incorporating a previous time step $\theta_{k-1}$, and rearrange into the form of a finite difference discretization, as shown in equation (**??**). Like all discretizations, the Euler discretizations introduce error terms proportional to the step size, which in this case is the learning rate $\eta$. Taylor expanding $\theta_{k+1}$ and $\theta_{k-1}$ around $\theta_k$, its easy to show that both Euler discretizations introduce a second-order error term proportional to $\frac{\eta}{2}\ddot{\theta}$.

$$\frac{\theta_{k+1} - \theta_k}{\eta} = \dot{\theta} + \frac{\eta}{2}\ddot{\theta} + O(\eta^2), \qquad \frac{\theta_k - \theta_{k-1}}{\eta} = \dot{\theta} - \frac{\eta}{2}\ddot{\theta} + O(\eta^2).$$

Notice how the momentum coefficient $\beta \in [0, 1]$ regulates the amount of backward Euler incorporated into the discretization. When $\beta = 0$, we remove all backward Euler discretization leaving just the forward Euler discretization. When $\beta = 1$, we have equal amounts of backward Euler as forward Euler resulting in a central second-order discretization[2] as noticed in [19].

## A.2  MODELING STOCHASTICITY

In order to model the effect of stochastic batches, we first model a batch gradient with the following assumption:

**Assumption 1** (CLT). *We assume the batch gradient is a noisy version of the true gradient such that $g_\mathcal{B}(\theta) - g(\theta)$ is a Gaussian random variable with mean $0$ and covariance $\frac{1}{S}\Sigma(\theta)$.*

The two conditions needed for the CLT to hold are not exactly met in the setting of SGD. *Independent and identically distributed.* Generally we perform SGD by making a complete pass through the entire dataset before using a sample again which introduces a weak dependence between samples. While the covariance matrix without replacement more accurately models the dependence between samples within a batch, it fails to account for the dependence between batches. *Finite variance.* A different line of work has questioned the Gaussian assumption entirely because of the need for finite variance random variables. This work instead suggests using the generalized central limit theorem implying the noise would be a heavy-tailed $\alpha$-stable random variable [29]. Thus, the previous assumption is implicitly assuming the *i.i.d.* and finite variance conditions apply for large enough datasets and small enough batches.

Under the CLT assumption, we must also replace the Euler discretizations with Euler–Maruyama discretizations. For a general stochastic process, $dX_t = \mu dt + \sigma dW_t$, the Euler–Maruyama method extends the Euler method for ODEs to SDEs, resulting in the update equation $X_{k+1} = X_k + \Delta t\mu + \sqrt{\Delta t}\sigma\xi$, where $\xi \sim \mathcal{N}(0, 1)$. Notice, the key difference is that if the temporal step size is $\Delta t = \eta$, then the noise is scaled by the square root $\sqrt{\eta}$. In fact, the main argument against modeling SGD with an SDE, as nicely explained in Yaida [28], is that most SDE approximations simultaneously assume that $\Delta t \to 0^+$, while maintaining that the square root of the learning rate $\sqrt{\eta}$ is finite. However, by modeling the discretization and stochastic effect simultaneously we can avoid this argument, bringing us to our second assumption:

**Assumption 2** (SDE). *We assume the underdamped Langevin equation (3) accurately models the trajectory of the network driven by SGD through phase space such that $\theta(\eta k) \approx \theta_k$ and $v(\eta k) \approx v_k$.*

This approach of modeling discretization and stochasticity simultaneously is called stochastic modified equations, as further explained in Li et al. [22].

---

[2]The difference between a forward Euler and backward Euler discretization is a second-order central discretization, $\left(\frac{\theta_{k+1}-\theta_k}{\eta}\right) - \left(\frac{\theta_k-\theta_{k-1}}{\eta}\right) = \eta\left(\frac{\theta_{k+1}-2\theta_k+\theta_{k-1}}{\eta^2}\right) = \eta\ddot{\theta} + O(\eta^2)$.

## B    STRUCTURE IN THE COVARIANCE OF THE GRADIENT NOISE

As we've mentioned before, SGD introduces highly structured noise into an optimization process, often assumed to be an essential ingredient for its ability to avoid local minima.

**Assumption 5** (Covariance Structure). *We assume the covariance of the gradient noise is proportional to the Hessian of the quadratic loss $\Sigma(\theta) = \sigma^2 H$ where $\sigma \in \mathbb{R}^+$ is some unknown scalar.*

In the setting of linear regression, this is a very natural assumption. If we assume the classic generative model for linear regression data $y_i = x_i^\intercal \bar{\theta} + \sigma \epsilon$ where, $\bar{\theta} \in \mathbb{R}^d$ is the true model and $\epsilon \sim \mathcal{N}(0, 1)$, then provably $\Sigma(\theta) \approx \sigma^2 H$.

*Proof.* We can estimate the covariance as $\Sigma(\theta) \approx \frac{1}{N} \sum_{i=1}^{N} g_i g_i^\intercal - gg^\intercal$. Near stationarity $gg^\intercal \ll \frac{1}{N} \sum_{i=1}^{N} g_i g_i^\intercal$, and thus,

$$\Sigma(\theta) \approx \frac{1}{N} \sum_{i=1}^{N} g_i g_i^\intercal.$$

Under the generative model $y_i = x_i^\intercal \bar{\theta} + \sigma \epsilon$ where $\epsilon \sim \mathcal{N}(0, 1)$ and $\sigma \in \mathbb{R}^+$, then the gradient $g_i$ is

$$g_i = (x_i^\intercal (\theta - \bar{\theta}) - \sigma \epsilon) x_i,$$

and the matrix $g_i g_i^\intercal$ is

$$g_i g_i^\intercal = (x_i^\intercal (\theta - \bar{\theta}) - \sigma \epsilon)^2 (x_i x_i^\intercal).$$

Assuming $\theta \approx \bar{\theta}$ at stationarity, then $(x_i^\intercal (\theta - \bar{\theta}) - \sigma \epsilon)^2 \approx \sigma^2$. Thus,

$$\Sigma(\theta) \approx \frac{\sigma^2}{N} \sum_{i=1}^{N} x_i x_i^\intercal = \frac{\sigma^2}{N} X^\intercal X = \sigma^2 H$$

Also notice that weight decay is independent of the data or batch and thus simply shifts the gradient distribution, but leaves the covariance of the gradient noise unchanged. □

While the above analysis is in the linear regression setting, for deep neural networks it is reasonable to make the same assumption. See the appendix of Jastrzębski et al. [12] for a discussion on this assumption in the non-linear setting.

Recent work by Ali et al. [32] also studies the dynamics of SGD (without momentum) in the setting of linear regression. This work, while studying the classic first-order stochastic differential equation, made a point to not introduce an assumption on the diffusion matrix. In particular, they make the point that even in the setting of linear regression, a constant covariance matrix will fail to capture the actual dynamics. To illustrate this point they consider the univariate responseless least squares problem,

$$\underset{\theta \in \mathbb{R}}{\text{minimize}} \quad \frac{1}{2n} \sum_{i=1}^{n} (x_i \theta)^2.$$

As they explain, the SGD update for this problem would be

$$\theta_{k+1} = \theta_k - \frac{\eta}{S} \left( \sum_{i \in \mathcal{B}} x_i \right) \theta_k = \prod_{i=1}^{k} (1 - \eta(\tfrac{1}{S} \sum_{i \in \mathcal{B}} x_i)) \theta_0,$$

from which they conclude for a small enough learning rate $\eta$, then with probability one $\theta_k \to 0$. They contrast this with the Ornstein-Uhlenbeck process given by a constant covariance matrix where while the mean for $\theta_k$ converges to zero its variance converges to a positive constant. So is this discrepancy evidence that an Ornstein-Uhlenbeck process with a constant covariance matrix fails to capture the updates of SGD? In many ways this problem is not a simple example, rather a pathological edge case. Consider the generative model that would give rise to this problem,

$$y = 0x + 0\xi = 0.$$

In otherwords, the true model $\bar{\theta} = 0$ and the standard deviation for the noise $\sigma = 0$. This would imply by the assumption used in our paper that there would be zero diffusion and the resulting SDE would simplify to a deterministic ODE that exponentially converges to zero.

## C    A QUADRATIC LOSS AT THE END OF TRAINING

**Assumption 4** (Quadratic Loss). *We assume that at the end of training the loss for a neural network can be approximated by the quadratic loss* $\mathcal{L}(\theta) = (\theta - \mu)^{\intercal} \left( \frac{H}{2} \right) (\theta - \mu)$, *where* $H \succeq 0$ *is the training loss Hessian and* $\mu$ *is some unknown mean vector, corresponding to a local minimum.*

This assumption has been amply used in previous works such as Mandt et al. [31], Jastrzębski et al. [12], and Poggio et al. [50]. Particularly, Mandt et al. [31] discuss how this assumption makes sense for smooth loss functions for which the stationary solution to the stochastic process reaches a deep local minimum from which it is difficult to escape.

It is a well-studied fact, both empirically and theoretically, that the Hessian is low-rank near local minima as noted by Sagun et al. [51], and Kunin et al. [20]. This degeneracy results in flat directions of equal loss. Kunin et al. [20] discuss how differentiable symmetries, architectural features that keep the loss constant under certain weight transformations, give rise to these flat directions. Importantly, the Hessian and the covariance matrix share the same null space, and thus we can always restrict ourselves to the image space of the Hessian, where the drift and diffusion matrix will be full rank. Further discussion on the relationship between the Hessian and the covariance matrix can be found in Thomas et al. [52].

It is also a well known empirical fact that even at the end of training the Hessian can have negative eigenvalues [41]. This empirical observation is at odds with our assumption that the Hessian is positive semi-definite $H \succeq 0$. Further analysis is needed to alleviate this inconsistency.

## D  Solving an Ornstein-Uhlenbeck Process with Anisotropic Noise

We will study the multivariate Ornstein-Uhlenbeck process described by the stochastic differential equation

$$dX_t = A(\mu - X_t)dt + \sqrt{2\kappa^{-1}D}dW_t \qquad X_0 = x_0, \tag{14}$$

where $A \in \mathbb{S}_{++}^m$ is a positive definite drift matrix, $\mu \in \mathbb{R}^m$ is a mean vector, $\kappa \in \mathbb{R}^+$ is some positive constant, and $D \in \mathbb{S}_{++}^m$ is a positive definite diffusion matrix. This OU process is unique in that it is one of the few SDEs we can solve explicitly. We can derive an expression for $X_T$ as,

$$X_T = e^{-AT}x_0 + \left(I - e^{-AT}\right)\mu + \int_0^T e^{A(t-T)}\sqrt{2\kappa^{-1}D}dW_t. \tag{15}$$

*Proof.* Consider the function $f(t, x) = e^{At}x$ where $e^A$ is a matrix exponential. Then by Itô's Lemma[3] we can evaluate the derivative of $f(t, X_t)$ as

$$df(t, X_t) = \left(Ae^{At}X_t + e^{At}A(\mu - X_t)\right)dt + e^{At}\sqrt{2\kappa^{-1}D}dW_t$$
$$= Ae^{At}\mu dt + e^{At}\sqrt{2\kappa^{-1}D}dW_t$$

Integrating this expression from $t = 0$ to $t = T$ gives

$$f(T, X_T) - f(0, X_0) = \int_0^T Ae^{At}\mu dt + \int_0^T e^{At}\sqrt{2\kappa^{-1}D}dW_t$$

$$e^{AT}X_T - x_0 = \left(e^{AT} - I\right)\mu + \int_0^T e^{At}\sqrt{2\kappa^{-1}D}dW_t$$

which rearranged gives the expression for $X_T$. □

From this expression it is clear that $X_T$ is a Gaussian process. The mean of the process is

$$\mathbb{E}\left[X_T\right] = e^{-AT}x_0 + \left(I - e^{-AT}\right)\mu, \tag{16}$$

and the covariance and cross-covariance of the process are

$$\text{Var}(X_T) = \kappa^{-1} \int_0^T e^{A(t-T)}2De^{A^\intercal(t-T)}dt, \tag{17}$$

$$\text{Cov}(X_T, X_S) = \kappa^{-1} \int_0^{\min(T,S)} e^{A(t-T)}2De^{A^\intercal(t-S)}dt. \tag{18}$$

These last two expressions are derived by Itô Isometry[4].

### D.1  The Lyapunov Equation

We can explicitly solve the integral expressions for the covariance and cross-covariance exactly by solving for the unique matrix $B \in \mathbb{S}_{++}^m$ that solves the *Lyapunov equation*,

$$AB + BA^\intercal = 2D. \tag{19}$$

If $B$ solves the Lyapunov equation, notice

$$\frac{d}{dt}\left(e^{A(t-T)}Be^{A^\intercal(t-S)}\right) = e^{A(t-T)}ABe^{A^\intercal(t-S)} + e^{A(t-T)}BA^\intercal e^{A^\intercal(t-S)}$$
$$= e^{A(t-T)}2De^{A^\intercal(t-S)}$$

Using this derivative, the integral expressions for the covariance and cross-covariance simplify as,

$$\text{Var}(X_T) = \kappa^{-1}\left(B - e^{-AT}Be^{-A^\intercal T}\right), \tag{20}$$

$$\text{Cov}(X_T, X_S) = \kappa^{-1}\left(B - e^{-AT}Be^{-A^\intercal T}\right)e^{A^\intercal(T-S)}, \tag{21}$$

where we implicitly assume $T \le S$.

---

[3]Itô's Lemma states that for any Itô drift-diffusion process $dX_t = \mu_t dt + \sigma_t dW_t$ and twice differentiable scalar function $f(t, x)$, then $df(t, X_t) = \left(f_t + \mu_t f_x + \frac{\sigma_t^2}{2}f_{xx}\right)dt + \sigma_t f_x dW_t$.

[4]Itô Isometry states for any standard Itô process $X_t$, then $\mathbb{E}\left[\left(\int_0^t X_t dW_t\right)^2\right] = \mathbb{E}\left[\int_0^t X_t^2 dt\right]$.

## D.2 DECOMPOSING THE DRIFT MATRIX

While the Lyapunov equation simplifies the expressions for the covariance and cross-covariance, it does not explain how to actually solve for the unknown matrix $B$. Following a method proposed by Kwon et al. [48], we will show how to solve for $B$ explicitly in terms of the drift $A$ and diffusion $D$.

The drift matrix $A$ can be uniquely decomposed as,
$$A = (D + Q)U \tag{22}$$
where $D$ is our symmetric diffusion matrix, $Q$ is a skew-symmetric matrix (i.e. $Q = -Q^{\mathsf{T}}$), and $U$ is a positive definite matrix. Using this decomposition, then $B = U^{-1}$, solves the Lyapunov equation.

*Proof.* Plug $B = U^{-1}$ into the left-hand side of equation (19),
$$\begin{aligned} AU^{-1} + U^{-1}A^{\mathsf{T}} &= (D+Q)UU^{-1} + U^{-1}U(D-Q) \\ &= (D+Q) + (D-Q) \\ &= 2D \end{aligned}$$
Here we used the symmetry of $A, D, U$ and the skew-symmetry of $Q$. □

All that is left is to do is solve for the unknown matrices $Q$ and $U$. First notice the following identity,
$$AD - DA = QA + AQ \tag{23}$$

*Proof.* Multiplying $A = (D+Q)U$ on the right by $(D-Q)$ gives,
$$\begin{aligned} A(D-Q) &= (D+Q)U(D-Q) \\ &= (D+Q)A^{\mathsf{T}}, \end{aligned}$$
which rearranged and using $A = A^{\mathsf{T}}$ gives the desired equation. □

Let $V\Lambda V^{\mathsf{T}}$ be the eigendecomposition of $A$ and define the matrices $\widetilde{D} = V^{\mathsf{T}}DV$ and $\widetilde{Q} = V^{\mathsf{T}}QV$. These matrices observe the following relationship,
$$\widetilde{Q}_{ij} = \frac{\lambda_i - \lambda_j}{\rho_i + \lambda_j}\widetilde{D}_{ij}. \tag{24}$$

*Proof.* Replace $A$ in the previous equality with its eigendecompsoition,
$$V\Lambda V^{\mathsf{T}}D - DV\Lambda V^{\mathsf{T}} = QV\Lambda V^{\mathsf{T}} + V\Lambda V^{\mathsf{T}}Q.$$
Multiply this equation on the right by $V$ and on the left by $V^{\mathsf{T}}$,
$$\Lambda\widetilde{D} - \widetilde{D}\Lambda = \widetilde{Q}\Lambda + \Lambda\widetilde{Q}.$$
Looking at this equality element-wise and using the fact that $\Lambda$ is diagonal gives the scalar equality for any $i, j$,
$$(\lambda_i - \lambda_j)\widetilde{D}_{ij} = (\lambda_i + \lambda_j)\widetilde{Q}_{ij},$$
which rearranged gives the desired expression. □

Thus, $Q$ and $U$ are given by,
$$Q = V\widetilde{Q}V^{\mathsf{T}}, \qquad U = (D+Q)^{-1}A. \tag{25}$$
This decomposition always holds uniquely when $A, D \succ 0$, as $\frac{\lambda_i - \lambda_j}{\lambda_i + \lambda_j}$ exists and $(D+Q)$ is invertible. See [48] for a discussion on the singularities of this decomposition.

## D.3 STATIONARY SOLUTION

Using the Lyapunov equation and the drift decomposition, then $X_T \sim p_T$, where
$$p_T = \mathcal{N}\left(e^{-AT}x_0 + \left(I - e^{-AT}\right)\mu, \kappa^{-1}\left(U^{-1} - e^{-AT}U^{-1}e^{-A^{\mathsf{T}}T}\right)\right). \tag{26}$$
In the limit as $T \to \infty$, then $e^{-AT} \to 0$ and $p_T \to p_{ss}$ where
$$p_{ss} = \mathcal{N}\left(\mu, \kappa^{-1}U^{-1}\right). \tag{27}$$
Similarly, the cross-covariance converges to the stationary cross-covariance,
$$\mathrm{Cov}_{ss}(X_T, X_S) = \kappa^{-1}Be^{A^{\mathsf{T}}(T-S)}. \tag{28}$$

# E   A VARIATIONAL FORMULATION OF THE OU PROCESS WITH ANISOTROPIC NOISE

In this section we will describe an alternative, variational, route towards solving the dynamics of the OU process studied in appendix D.

Let $\Phi : \mathbb{R}^n \to \mathbb{R}$ be an arbitrary, non-negative potential and consider the stochastic differential equation describing the Langevin dynamics of a particle in this potential field,

$$dX_t = -\nabla \Phi(X_t)dt + \sqrt{2\kappa^{-1}D(X_t)}dW_t, \qquad X_0 = x_0, \tag{29}$$

where $D(X_t)$ is an arbitrary, spatially-dependent, diffusion matrix, $\kappa$ is a temperature constant, and $x_0 \in \mathbb{R}^m$ is the particle's initial position. The *Fokker-Planck equation* describes the time evolution for the probability distribution $p$ of the particle's position such that $p(x, t) = \mathbb{P}(X_t = x)$. The FP equation is the partial differential equation[5],

$$\partial_t p = \nabla \cdot \left( \nabla \Phi(X_t)p + \kappa^{-1}\nabla \cdot (D(X_t)p) \right), \qquad p(x, 0) = \delta(x_0), \tag{30}$$

where $\nabla\cdot$ denotes the divergence and $\delta(x_0)$ is a dirac delta distribution centered at the initialization $x_0$. To assist in the exploration of the FP equation we define the vector field,

$$J(x, t) = -\nabla \Phi(X_t)p - \nabla \cdot (D(X_t)p), \tag{31}$$

which is commonly referred to as the *probability current*. Notice, that this gives an alternative expression for the FP equation, $\partial_t p = -\nabla \cdot J$, demonstrating that $J(x, t)$ defines the flow of probability mass through space and time. This interpretation is especially useful for solving for the *stationary solution* $p_{ss}$, which is the unique distribution that satisfies,

$$\partial_t p_{ss} = -\nabla \cdot J_{ss} = 0, \tag{32}$$

where $J_{ss}$ is the probability current for $p_{ss}$. The stationary condition can be obtained in two distinct ways:

1. *Detailed balance.* This is when $J_{ss}(x) = 0$ for all $x \in \Omega$. This is analogous to reversibility for discrete Markov chains, which implies that the probability mass flowing from a state $i$ to any state $j$ is the same as the probability mass flowing from state $j$ to state $i$.

2. *Broken detailed balance.* This is when $\nabla \cdot J_{ss}(x) = 0$ but $J_{ss}(x) \neq 0$ for all $x \in \Omega$. This is analogous to irreversibility for discrete Markov chains, which only implies that the total probability mass flowing out of state $i$ equals to the total probability mass flowing into state $i$.

The distinction between these two cases is critical for understanding the limiting dynamics of the process.

## E.1   THE VARIATIONAL FORMULATION OF THE FOKKER-PLANCK EQUATION WITH ISOTROPIC DIFFUSION

We will now consider the restricted setting of standard, isotropic diffusion ($D = I$). It is easy enough to check that in this setting the stationary solution is

$$p_{ss}(x) = \frac{e^{-\kappa \Phi(x)}}{Z}, \qquad Z = \int_\Omega e^{-\kappa \Phi(x)}dx, \tag{33}$$

where $p_{ss}$ is called a *Gibbs distribution* and $Z$ is the partition function. Under this distribution, the stationary probability current is zero ($J_{ss}(x) = 0$) and thus the process is in detailed balance. Interestingly, the Gibbs distribution $p_{ss}$ has another interpretation as the unique minimizer of the the *Gibbs free energy* functional,

$$F(p) = \mathbb{E}\left[\Phi\right] - \kappa^{-1}H(p), \tag{34}$$

where $\mathbb{E}\left[\Phi\right]$ is the expectation of the potential $\Phi$ under the distribution $p$ and $H(p) = -\int_\Omega p(x)log(p(x))dx$ is the Shannon entropy of $p$.

---

[5]This PDE is also known as the Forward Kolmogorov equation.

*Proof.* To prove that indeed $p_{ss}$ is the unique minimizer of the Gibbs free energy functional, consider the following equivalent expression

$$
\begin{aligned}
F(p) &= \int_\Omega p(x)\Phi(x)dx + \kappa^{-1}\int_\Omega p(x)log(p(x))dx \\
&= \kappa^{-1}\int_\Omega p(x)\left(log(p(x)) - log(p_{ss}(x))\right)dx - \kappa^{-1}\int_\Omega log(Z) \\
&= \kappa^{-1}D_{\mathrm{KL}}(p \parallel p_{ss}) - \kappa^{-1}log(Z)
\end{aligned}
$$

From this expressions, it is clear that the Kullback–Leibler divergence is uniquely minimized when $p = p_{ss}$. $\square$

In other words, with isotropic diffusion the stationary solution $p_{ss}$ can be thought of as the limiting distribution given by the Fokker-Planck equation or the unique minimizer of an energetic-entropic functional.

Seminal work by Jordan et al. [53] deepened this connection between the Fokker-Planck equation and the Gibbs free energy functional. In particular, their work demonstrates that the solution $p(x, t)$ to the Fokker-Planck equation is the *Wasserstein gradient flow* trajectory on the Gibbs free energy functional.

Steepest descent is always defined with respect to a distance metric. For example, the update equation, $x_{k+1} = x_k - \eta\nabla\Phi(x_k)$, for classic gradient descent on a potential $\Phi(x)$, can be formulated as the solution to the minimization problem $x_{k+1} = \operatorname{argmin}_x \eta\Phi(x) + \frac{1}{2}d(x, x_k)^2$ where $d(x, x_k) = \|x - x_k\|$ is the Euclidean distance metric. Gradient flow is the continuous-time limit of gradient descent where we take $\eta \to 0^+$. Similarly, Wasserstein gradient flow is the continuous-time limit of steepest descent optimization defined by the *Wasserstein metric*. The Wasserstein metric is a distance metric between probability measures defined as,

$$
W_2^2(\mu_1, \mu_2) = \inf_{p \in \Pi(\mu_1, \mu_2)} \int_{\mathbb{R}^n \times \mathbb{R}^n} |x - y|^2 p(dx, dy), \tag{35}
$$

where $\mu_1$ and $\mu_2$ are two probability measures on $\mathbb{R}^n$ with finite second moments and $\Pi(\mu_1, \mu_2)$ defines the set of joint probability measures with marginals $\mu_1$ and $\mu_2$. Thus, given an initial distribution and learning rate $\eta$, we can use the Wasserstein metric to derive a sequence of distributions minimizing some functional in the sense of steepest descent. In the continuous-time limit as $\eta \to 0^+$ this sequence defines a continuous trajectory of probability distributions minimizing the functional. Jordan et al. [54] proved, through the following theorem, that this process applied to the Gibbs free energy functional converges to the solution to the Fokker-Planck equation with the same initialization:

**Theorem 1** (JKO). *Given an initial condition $p_0$ with finite second moment and an $\eta > 0$, define the iterative scheme $p_\eta$ with iterates defined by*

$$
p_k = \operatorname{argmin}_p \eta\left(\mathbb{E}\left[\Phi\right] - \kappa^{-1}H(p)\right) + W_2^2(p, p^{k-1}).
$$

*As $\eta \to 0^+$, then $p_\eta \to p$ weakly in $L^1$ where $p$ is the solution to the Fokker-Planck equation with the same initial condition.*

See [54] for further explanation and [53] for a complete derivation.

### E.2 EXTENDING THE VARIATIONAL FORMULATION TO THE SETTING OF ANISOTROPIC DIFFUSION

While the JKO theorem provides a very powerful lens through which to view solutions to the Fokker-Planck equation, and thus distributions for particles governed by Langevin dynamics, it only applies in the very restricted setting of isotropic diffusion. In this section we will review work by Chaudhari and Soatto [33] extending the variational interpretation to the setting of anisotropic diffusion.

Consider when $D(X_t)$ is an anisotropic, spatially-dependent diffusion matrix. In this setting, the original Gibbs distribution given in equation (33) does not necessarily satisfy the stationarity condition equation (32). In fact, it is not immediately clear what the stationary solution is or if the dynamics even have one. Thus, Chaudhari and Soatto [33] make the following assumption:

**Stationary Assumption.** *Assume there exists a unique distribution $p_{ss}$ that is the stationary solution to the Fokker-Planck equation irregardless of initial conditions.*

Under this assumption we can implicitly define the potential $\Psi(x) = -\kappa^{-1}log(p_{ss}(x))$. Using this modified potential we can express the stationary solution as a Gibbs distribution,

$$p_{ss}(x) \propto e^{-\kappa\Psi(x)}. \tag{36}$$

Under this implicit definition we can define the stationary probability current as $J_{ss}(x) = j(x)p_{ss}(x)$ where

$$j(x) = -\nabla\Phi(x) - \kappa^{-1}\nabla \cdot D(x) + D(x)\nabla\Psi(x). \tag{37}$$

The vector field $j(x)$ reflects the discrepancy between the original potential $\Phi$ and the modified potential $\Psi$ according to the diffusion $D(x)$. Notice that in the isotropic case, when $D(x) = I$, then $\Phi = \Psi$ and $j(x) = 0$. Chaudhari and Soatto [33] introduce another property of $j(x)$ through assumption,

**Conservative Assumption.** *Assume that the force $j(x)$ is conservative (i.e. $\nabla \cdot j(x) = 0$).*

Using this assumption, Chaudhari and Soatto [33] extends the variational formulation provided by the JKO theorem to the anisotropic setting,

**Theorem 2** (CS)**.** *Given an initial condition $p_0$ with finite second moment, then the energetic-entropic functional,*

$$F(p) = \mathbb{E}_p\left[\Psi(x)\right] - \kappa^{-1}H(p)$$

*monotonically decreases throughout the trajectory given by the solution to the Fokker-Planck equation with the given initial condition.*

In other words, the Fokker-Plank equation (30) with anisotropic diffusion can be interpreted as minimizing the expectation of a modified loss $\Psi$, while being implicitly regularized towards distributions that maximize entropy. The derivation requires we assume a stationary solution $p_{ss}$ exists and that the force $j(x)$ implicitly defined by $p_{ss}$ is conservative. However, rather than implicitly define $\Psi(x)$ and $j(x)$ through assumption, if we can explicitly construct a modified loss $\Psi(x)$ such that the resulting $j(x)$ satisfies certain conditions, then the stationary solution exists and the variational formulation will apply as well. We formalize this statement with the following theorem,

**Theorem 3** (Explicit Construction)**.** *If there exists a potential $\Psi(x)$ such that either $j(x) = 0$ or $\nabla \cdot j(x) = 0$ and $\nabla\Psi(x) \perp j(x)$, then $p_{ss}$ is the Gibbs distribution $\propto e^{-\kappa\Psi(x)}$ and the variational formulation given in Theorem 2 applies.*

### E.3 Applying the Variational Formulation to the OU Process

Through explicit construction we now seek to find analytic expressions for the modified loss $\Psi(x)$ and force $j(x)$ hypothesised by Chaudhari and Soatto [33] in the fundamental setting of an OU process with anisotropic diffusion, as described in section D. We assume the diffusion matrix is anisotropic, but spatially independent, $\nabla \cdot D(x) = 0$. For the OU process the original potential generating the drift is

$$\Phi(x) = (x - \mu)^\mathsf{T}\frac{A}{2}(x - \mu). \tag{38}$$

Recall, that in order to extend the variational formulation we must construct some potential $\Psi(x)$ such that $\nabla \cdot j(x) = 0$ and $\nabla\Psi \perp j(x)$. It is possible to construct $\Psi(x)$ using the unique decomposition of the drift matrix $A = (D + Q)U$ discussed in appendix D. Define the modified potential,

$$\Psi(x) = (x - \mu)^\mathsf{T}\frac{U}{2}(x - \mu). \tag{39}$$

Using this potential, the force $j(x)$ is

$$j(x) = -A(x - \mu) + DU(x - \mu) = -QU(x - \mu). \tag{40}$$

Notice that $j(x)$ is conservative, $\nabla \cdot j(x) = \nabla \cdot -QU(x - \mu) = 0$ because $Q$ is skew-symmetric. Additionally, $j(x)$ is orthogonal, $j(x)^\mathsf{T}\nabla\Psi(x) = (x - \mu)^\mathsf{T}U^\mathsf{T}QU(x - \mu) = 0$, again because $Q$ is skew-symmetric. Thus, we have determined a modified potential $\Psi(x)$ that results in a conservative orthogonal force $j(x)$ satisfying the conditions for Theorem 3. Indeed the stationary Gibbs distribution given by Theorem 3 agrees with equation (27) derived via the first and second moments in appendix D,

$$e^{-\kappa\Psi(x)} \propto \mathcal{N}\left(\mu, \kappa^{-1}U^{-1}\right)$$

In addition to the variational formulation, this interpretation further details explicitly the stationary probability current, $J_{ss}(x) = j(x)p_{ss}$, and whether or not the the stationary solution is in broken detailed balance.

## F   EXPLICIT EXPRESSIONS FOR THE OU PROCESS GENERATED BY SGD

We will now consider the specific OU process generated by SGD with linear regression. Here we repeat the setup as explained in section 5.

Let $X \in \mathbb{R}^{N \times d}$, $Y \in \mathbb{R}^N$ be the input data, output labels respectively and $\theta \in \mathbb{R}^d$ be our vector of regression coefficients. The least squares loss is the convex quadratic loss $\mathcal{L}(\theta) = \frac{1}{2N}\|Y - X\theta\|^2$ with gradient $g(\theta) = H\theta - b$, where $H = \frac{X^\intercal X}{N}$ and $b = \frac{X^\intercal Y}{N}$. Plugging this expression for the gradient into the underdamped Langevin equation (3), and rearranging terms, results in the multivariate Ornstein-Uhlenbeck (OU) process,

$$d \begin{bmatrix} \theta_t \\ v_t \end{bmatrix} = A \left( \begin{bmatrix} \mu \\ 0 \end{bmatrix} - \begin{bmatrix} \theta_t \\ v_t \end{bmatrix} \right) dt + \sqrt{2\kappa^{-1}D}dW_t, \tag{41}$$

where $A$ and $D$ are the drift and diffusion matrices respectively,

$$A = \begin{bmatrix} 0 & -I \\ \frac{2}{\eta(1+\beta)}(H + \lambda I) & \frac{2(1-\beta)}{\eta(1+\beta)}I \end{bmatrix}, \qquad D = \begin{bmatrix} 0 & 0 \\ 0 & \frac{2(1-\beta)}{\eta(1+\beta)}\Sigma(\theta) \end{bmatrix}, \tag{42}$$

$\kappa = S(1 - \beta^2)$ is a temperature constant, and $\mu = (H + \lambda I)^{-1}b$ is the ridge regression solution.

### F.1   SOLVING FOR THE MODIFIED LOSS AND CONSERVATIVE FORCE

In order to apply the expressions derived for a general OU process in appendix D and E, we must first decompose the drift as $A = (D + Q)U$. Under the simplification $\Sigma(\theta) = \sigma^2 H$ discussed in appendix B, then the matrices $Q$ and $U$, as defined below, achieve this,

$$Q = \begin{bmatrix} 0 & -\sigma^2 H \\ \sigma^2 H & 0 \end{bmatrix}, \qquad U = \begin{bmatrix} \frac{2}{\eta(1+\beta)\sigma^2}H^{-1}(H + \lambda I) & 0 \\ 0 & \frac{1}{\sigma^2}H^{-1} \end{bmatrix}. \tag{43}$$

Using these matrices we can now derive explicit expressions for the modified loss $\Psi(\theta, v)$ and conservative force $j(\theta, v)$. First notice that the least squares loss with $L_2$ regularization is proportional to the convex quadratic,

$$\Phi(\theta) = (\theta - \mu)^\intercal (H + \lambda I)(\theta - \mu). \tag{44}$$

The modified loss $\Psi$ is composed of two terms, one that only depends on the position,

$$\Psi_\theta(\theta) = (\theta - \mu)^\intercal \left( \frac{H^{-1}(H + \lambda I)}{\eta(1 + \beta)\sigma^2} \right) (\theta - \mu), \tag{45}$$

and another that only depends on the velocity,

$$\Psi_v(v) = v^\intercal \left( \frac{H^{-1}}{\sigma^2} \right) v. \tag{46}$$

The conservative force $j(\theta, v)$ is

$$j(\theta, v) = \begin{bmatrix} v \\ -\frac{2}{\eta(1+\beta)}(H + \lambda I)(\theta - \mu) \end{bmatrix}, \tag{47}$$

and thus the stationary probability current is $J_{ss}(\theta, v) = j(\theta, v)p_{ss}$.

### F.2   DECOMPOSING THE TRAJECTORY INTO THE EIGENBASIS OF THE HESSIAN

As shown in appendix D, the temporal distribution for the OU process at some time $T \geq 0$ is,

$$p_T \left( \begin{bmatrix} \theta \\ v \end{bmatrix} \right) = \mathcal{N} \left( e^{-AT} \begin{bmatrix} \theta_0 \\ v_0 \end{bmatrix} + (I - e^{-AT}) \begin{bmatrix} \mu \\ 0 \end{bmatrix}, \kappa^{-1} \left( U^{-1} - e^{-AT}U^{-1}e^{-A^\intercal T} \right) \right).$$

Here we will now use the eigenbasis $\{q_1, \ldots, q_m\}$ of the Hessian with eigenvalues $\{\rho_1, \ldots, \rho_m\}$ to derive explicit expressions for the mean and covariance of the process through time.

**Deterministic component.** We can rearrange the expectation as

$$\mathbb{E}\left[\begin{bmatrix}\theta \\ v\end{bmatrix}\right] = \begin{bmatrix}\mu \\ 0\end{bmatrix} + e^{-AT}\begin{bmatrix}\theta_0 - \mu \\ v_0\end{bmatrix}.$$

Notice that the second, time-dependent term is actually the solution to the system of ODEs

$$\begin{bmatrix}\dot\theta \\ v\end{bmatrix} = -A\begin{bmatrix}\theta \\ v\end{bmatrix}$$

with initial condition $[\theta_0 - \mu \quad v_0]^\mathsf{T}$. This system of ODEs can be block diagonalized by factorizing $A = OSO^\mathsf{T}$ where $O$ is orthogonal and $S$ is block diagonal defined as

$$O = \begin{bmatrix} q_1 & 0 & \ldots & q_m & 0 \\ & \ddots & & & \\ 0 & q_1 & \ldots & 0 & q_m \end{bmatrix} \qquad S = \begin{bmatrix} 0 & -1 & & \\ \frac{2}{\eta(1+\beta)}(\rho_1 + \lambda) & \frac{2(1-\beta)}{\eta(1+\beta)} & & \ddots \\ & & \ddots & & \ddots \\ & & & 0 & -1 \\ & & \frac{2}{\eta(1+\beta)}(\rho_m + \lambda) & \frac{2(1-\beta)}{\eta(1+\beta)} \end{bmatrix}$$

In otherwords in the plane spanned by $[q_i \quad 0]^\mathsf{T}$ and $[0 \quad q_i]^\mathsf{T}$ the system of ODEs decouples into the 2D system

$$\begin{bmatrix}\dot a_i \\ b_i\end{bmatrix} = \begin{bmatrix} 0 & 1 \\ -\frac{2}{\eta(1+\beta)}(\rho_i + \lambda) & -\frac{2(1-\beta)}{\eta(1+\beta)} \end{bmatrix}\begin{bmatrix}a_i \\ b_i\end{bmatrix}$$

This system has a simple physical interpretation as a damped harmonic oscillator. If we let $b_i = \dot a_i$, then we can unravel this system into the second order ODE

$$\ddot a_i + 2\frac{1-\beta}{\eta(1+\beta)}\dot a_i + \frac{2}{\eta(1+\beta)}(\rho_i + \lambda)a_i = 0$$

which is in standard form (i.e. $\ddot x + 2\gamma\dot x + \omega^2 x = 0$) for $\gamma = \frac{1-\beta}{\eta(1+\beta)}$ and $\omega_i = \sqrt{\frac{2}{\eta(1+\beta)}(\rho_i + \lambda)}$. Let $a_i(0) = \langle\theta_0 - \mu, q_i\rangle$ and $b_i(0) = \langle v_0, q_i\rangle$, then the solution in terms of $\gamma$ and $\omega_i$ is

$$a_i(t) = \begin{cases} e^{-\gamma t}\left(a_i(0)\cosh\left(\sqrt{\gamma^2 - \omega_i^2}t\right) + \frac{\gamma a_i(0)+b_i(0)}{\sqrt{\gamma^2-\omega_i^2}}\sinh\left(\sqrt{\gamma^2-\omega_i^2}t\right)\right) & \gamma > \omega_i \\ e^{-\gamma t}(a_i(0) + (\gamma a_i(0) + b_i(0))t) & \gamma = \omega_i \\ e^{-\gamma t}\left(a_i(0)\cos\left(\sqrt{\omega_i^2 - \gamma^2}t\right) + \frac{\gamma a_i(0)+b_i(0)}{\sqrt{\omega_i^2-\gamma^2}}\sin\left(\sqrt{\omega_i^2-\gamma^2}t\right)\right) & \gamma < \omega_i \end{cases}$$

Differentiating these equations gives us solutions for $b_i(t)$

$$b_i(t) = \begin{cases} e^{-\gamma t}\left(b_i(0)\cosh\left(\sqrt{\gamma^2 - \omega_i^2}t\right) - \frac{\omega_i^2 a_i(0)+\gamma b_i(0)}{\sqrt{\gamma^2-\omega_i^2}}\sinh\left(\sqrt{\gamma^2-\omega_i^2}t\right)\right) & \gamma > \omega_i \\ e^{-\gamma t}\left(b_i(0) - \left(\omega_i^2 a_i(0) + \gamma b_i(0)\right)t\right) & \gamma = \omega_i \\ e^{-\gamma t}\left(b_i(0)\cos\left(\sqrt{\omega_i^2 - \gamma^2}t\right) - \frac{\omega_i^2 a_i(0)+\gamma b_i(0)}{\sqrt{\omega_i^2-\gamma^2}}\sin\left(\sqrt{\omega_i^2-\gamma^2}t\right)\right) & \gamma < \omega_i \end{cases}$$

Combining all these results, we can now analytically decompose the expectation as the sum,

$$\mathbb{E}\left[\begin{bmatrix}\theta \\ v\end{bmatrix}\right] = \begin{bmatrix}\mu \\ 0\end{bmatrix} + \sum_{i=1}^{m}\left(a_i(t)\begin{bmatrix}q_i \\ 0\end{bmatrix} + b_i(t)\begin{bmatrix}0 \\ q_i\end{bmatrix}\right).$$

Intuitively, this equation describes a damped rotation (spiral) around the OLS solution in the planes defined by the the eigenvectors of the Hessian at a rate proportional to the respective eigenvalue.

**Stochastic component.** Using the previous block diagonal decomposition $A = OSO^\mathsf{T}$ we can simplify the variance as

$$\mathrm{Var}\left(\begin{bmatrix}\theta \\ v\end{bmatrix}\right) = \kappa^{-1}\left(U^{-1} - e^{-AT}U^{-1}e^{-A^\mathsf{T}T}\right)$$

$$= \kappa^{-1}\left(U^{-1} - e^{-OSO^\mathsf{T}T}U^{-1}e^{-OS^\mathsf{T}O^\mathsf{T}T}\right)$$

$$= \kappa^{-1}O\left(O^\mathsf{T}U^{-1}O - e^{-ST}(O^\mathsf{T}U^{-1}O)e^{-ST^\mathsf{T}}\right)O^\mathsf{T}$$

Interestingly, the matrix $O^\mathsf{T}U^{-1}O$ is also block diagonal,

$$O^\mathsf{T}U^{-1}O = O^\mathsf{T}\begin{bmatrix}\frac{\eta(1+\beta)\sigma^2}{2}(H+\lambda I)^{-1}H & 0 \\ 0 & \sigma^2 H\end{bmatrix}O = \begin{bmatrix}\frac{\eta(1+\beta)\sigma^2}{2}\frac{\rho_1}{\rho_1+\lambda} & & 0 & \\ & 0 & \sigma^2\rho_1 & \\ & & & \ddots \\ & & \ddots & \ddots \\ & & & \frac{\eta(1+\beta)\sigma^2}{2}\frac{\rho_m}{\rho_m+\lambda} & 0 \\ & & & 0 & \sigma^2\rho_m\end{bmatrix}$$

Thus, similar to the mean, we can simply consider the variance in each of the planes spanned by $[q_i \quad 0]^\mathsf{T}$ and $[0 \quad q_i]^\mathsf{T}$. If we define the block matrices,

$$D_i = \begin{bmatrix}\frac{\eta\sigma^2}{2S(1-\beta)}\frac{\rho_i}{\rho_i+\lambda} & 0 \\ 0 & \frac{\sigma^2}{S(1-\beta^2)}\rho_i\end{bmatrix} \qquad S_i = \begin{bmatrix}0 & 1 \\ -\frac{2}{\eta(1+\beta)}(\rho_i+\lambda) & -\frac{2(1-\beta)}{\eta(1+\beta)}\end{bmatrix}$$

then the projected variance matrix in this plane simplifies as

$$\mathrm{Var}\left(\begin{bmatrix}q_i^\mathsf{T}\theta \\ q_i^\mathsf{T}v\end{bmatrix}\right) = D_i - e^{-S_iT}D_ie^{-S_iT^\mathsf{T}}$$

Using the solution to a damped harmonic osccilator discussed previously, we can express the matrix exponential $e^{-S_iT}$ explicitly in terms of $\gamma = \frac{1-\beta}{\eta(1+\beta)}$ and $\omega_i = \sqrt{\frac{2}{\eta(1+\beta)}(\rho_i+\lambda)}$. If we let $\alpha_i = \sqrt{|\gamma^2 - \omega_i^2|}$, then the matrix exponential is

$$e^{-S_it} = \begin{cases}e^{-\gamma t}\begin{bmatrix}\cosh(\alpha_i t) + \frac{\gamma}{\alpha_i}\sinh(\alpha_i t) & \frac{1}{\alpha_i}\sinh(\alpha_i t) \\ -\frac{\omega_i^2}{\alpha_i}\sinh(\alpha_i t) & \cosh(\alpha_i t) - \frac{\gamma}{\alpha_i}\sinh(\alpha_i t)\end{bmatrix} & \gamma > \omega_i \\[2em] e^{-\gamma t}\begin{bmatrix}1+\gamma t & t \\ -\omega_i^2 t & 1-\gamma t\end{bmatrix} & \gamma = \omega_i \\[2em] e^{-\gamma t}\begin{bmatrix}\cos(\alpha_i t) + \frac{\gamma}{\alpha_i}\sin(\alpha_i t) & \frac{1}{\alpha_i}\sin(\alpha_i t) \\ -\frac{\omega_i^2}{\alpha_i}\sin(\alpha_i t) & \cos(\alpha_i t) - \frac{\gamma}{\alpha_i}\sin(\alpha_i t)\end{bmatrix} & \gamma < \omega_i\end{cases}$$

## G   ANALYZING PROPERTIES OF THE STATIONARY SOLUTION

Assuming the stationary solution is given by equation (**??**) we can solve for the expected value of the norm of the local displacement and gain some intuition for the expected value of the norm of global displacement.

### G.1   INSTANTANEOUS SPEED

$$
\begin{aligned}
\mathbb{E}_{ss}\left[\|\delta_k\|^2\right] &= \mathbb{E}_{ss}\left[\|\theta_{k+1}-\theta_k\|^2\right] \\
&= \eta^2 \mathbb{E}_{ss}\left[\|v_{k+1}\|^2\right] \\
&= \eta^2 \text{tr}\left(\mathbb{E}_{ss}\left[v_{k+1}v_{k+1}^{\mathsf{T}}\right]\right) \\
&= \eta^2 \text{tr}\left(\text{Var}_{ss}\left(v_{k+1}\right)+\mathbb{E}_{ss}\left[v_{k+1}\right]\mathbb{E}_{ss}\left[v_{k+1}\right]^{\mathsf{T}}\right) \\
&= \eta^2 \text{tr}\left(\kappa^{-1}U^{-1}\right) \\
&= \frac{\eta^2}{S(1-\beta^2)}\text{tr}\left(\sigma^2 H\right)
\end{aligned}
$$

Note that this follows directly from the definition of $\delta_k$ in equation (1) and the mean and variance of the stationary solution in equation ( **??**), as well as the follow-up derivation in appendix F.

### G.2   ANOMALOUS DIFFUSION

Notice, that the global movement $\Delta_t = \theta_t - \theta_0$ can be broken up into the sum of the local movements $\Delta_t = \sum_{i=1}^{t}\delta_i$, where $\delta_i = \theta_i - \theta_{i-1}$. Applying this decomposition,

$$
\begin{aligned}
\mathbb{E}_{ss}\left[\|\Delta_t\|^2\right] &= \mathbb{E}_{ss}\left[\left\|\sum_{i=1}^{t}\delta_i\right\|^2\right] \\
&= \sum_{i=1}^{t}\mathbb{E}_{ss}\left[\|\delta_i\|^2\right]+\sum_{i\neq j}^{t}\mathbb{E}_{ss}\left[\langle\delta_i,\delta_j\rangle\right]
\end{aligned}
$$

As we solved for previously,

$$
\mathbb{E}_{ss}\left[\|\delta_i\|^2\right]=\eta^2\mathbb{E}_{ss}\left[\|v_i\|^2\right]=\eta^2\text{tr}\left(\text{Var}_{ss}(v_i)\right)=\frac{\eta^2}{S(1-\beta^2)}\text{tr}\left(\sigma^2 H\right).
$$

By a similar simplification, we can express the second term in terms of the stationary cross-covariance,

$$
\mathbb{E}_{ss}\left[\langle\delta_i,\delta_j\rangle\right]=\eta^2\mathbb{E}_{ss}\left[\langle v_i,v_j\rangle\right]=\eta^2\text{tr}\left(\text{Cov}_{ss}(v_i,v_j)\right).
$$

Thus, to simplify this expression we just need to consider the velocity-velocity covariance $\text{Cov}_{ss}(v_i,v_j)$. At stationarity, the cross-covariance for the system in phase space, $z_i=\begin{bmatrix}\theta_i & v_i\end{bmatrix}$ is

$$
\text{Cov}_{ss}(z_i,z_j)=\kappa^{-1}U^{-1}e^{-A^{\mathsf{T}}|i-j|}
$$

where $\kappa=S(1-\beta^2)$, and

$$
U=\begin{bmatrix}\frac{2}{\eta(1+\beta)\sigma^2}H^{-1}(H+\lambda I) & 0 \\ 0 & \frac{1}{\sigma^2}H^{-1}\end{bmatrix} \qquad A=\begin{bmatrix}0 & -I \\ \frac{2}{\eta(1+\beta)}(H+\lambda I) & \frac{2(1-\beta)}{\eta(1+\beta)}I\end{bmatrix}
$$

As discussed when solving for the mean of the OU trajectory, the drift matrix $A$ can be block diagonalized as $A=OSO^{\mathsf{T}}$ where $O$ is orthogonal and $S$ is block diagonal defined as

$$
O=\begin{bmatrix}q_1 & 0 & \dots & q_m & 0 \\ & & \ddots & & \\ 0 & q_1 & \dots & 0 & q_m\end{bmatrix}, \qquad S=\begin{bmatrix}0 & -1 & & \\ \frac{2}{\eta(1+\beta)}(\rho_1+\lambda) & \frac{2(1-\beta)}{\eta(1+\beta)} & \ddots & \\ & \ddots & \ddots & \\ & & 0 & -1 \\ & & \frac{2}{\eta(1+\beta)}(\rho_m+\lambda) & \frac{2(1-\beta)}{\eta(1+\beta)}\end{bmatrix}.
$$

Notice also that $O$ diagonalizes $U^{-1}$ such that,

$$\Lambda = O^{\mathsf{T}} U^{-1} O = \begin{bmatrix} \frac{\eta(1+\beta)\sigma^2}{2} \frac{\rho_1}{\rho_1+\lambda} & 0 & & & \\ 0 & \sigma^2 \rho_1 & \ddots & & \\ & \ddots & & \ddots & \\ & & & \frac{\eta(1+\beta)\sigma^2}{2} \frac{\rho_m}{\rho_m+\lambda} & 0 \\ & & & 0 & \sigma^2 \rho_m \end{bmatrix}.$$

Applying these decompositions, properties of matrix exponentials, and the cyclic invariance of the trace, allows us to express the trace of the cross-covariance as

$$\begin{aligned} \mathrm{tr}\left(\mathrm{Cov}_{ss}(z_i, z_j)\right) &= \kappa^{-1} \mathrm{tr}\left(U^{-1} e^{-A^{\mathsf{T}}|i-j|}\right) \\ &= \kappa^{-1} \mathrm{tr}\left(U^{-1} O e^{-S^{\mathsf{T}}|i-j|} O^{\mathsf{T}}\right) \\ &= \kappa^{-1} \mathrm{tr}\left(\Lambda e^{-S^{\mathsf{T}}|i-j|}\right) \\ &= \kappa^{-1} \sum_{k=1}^{n} \mathrm{tr}\left(\Lambda_k e^{-S_k^{\mathsf{T}}|i-j|}\right) \end{aligned}$$

where $\Lambda_k$ and $S_k$ are the blocks associated with each eigenvector of $H$. As solved for previously in the variance of the OU process, we can express the matrix exponential $e^{-S_k|i-j|}$ explicitly in terms of $\gamma = \frac{1-\beta}{\eta(1+\beta)}$ and $\omega_k = \sqrt{\frac{2}{\eta(1+\beta)}(\rho_k + \lambda)}$. If we let $\tau = |i-j|$ and $\alpha_k = \sqrt{|\gamma^2 - \omega_k^2|}$, then the matrix exponential is

$$e^{-S_k|i-j|} = \begin{cases} e^{-\gamma\tau} \begin{bmatrix} \cosh(\alpha_k\tau) + \frac{\gamma}{\alpha_k}\sinh(\alpha_k\tau) & \frac{1}{\alpha_k}\sinh(\alpha_k\tau) \\ -\frac{\omega_k^2}{\alpha_k}\sinh(\alpha_k\tau) & \cosh(\alpha_k\tau) - \frac{\gamma}{\alpha_k}\sinh(\alpha_k\tau) \end{bmatrix} & \gamma > \omega_k \\ e^{-\gamma\tau} \begin{bmatrix} 1 + \gamma\tau & \tau \\ -\omega_k^2\tau & 1 - \gamma\tau \end{bmatrix} & \gamma = \omega_k \\ e^{-\gamma\tau} \begin{bmatrix} \cos(\alpha_k\tau) + \frac{\gamma}{\alpha_k}\sin(\alpha_k\tau) & \frac{1}{\alpha_k}\sin(\alpha_k\tau) \\ -\frac{\omega_k^2}{\alpha_k}\sin(\alpha_k\tau) & \cos(\alpha_k\tau) - \frac{\gamma}{\alpha_k}\sin(\alpha_k\tau) \end{bmatrix} & \gamma < \omega_k \end{cases}$$

Plugging in these expressions into previous expression and restricting to just the $k^{\text{th}}$ velocity component, we see

$$\mathrm{Cov}_{ss}(v_{i,k}, v_{j,k}) = \kappa^{-1}\left[\Lambda_k e^{-S_k^{\mathsf{T}}|i-j|}\right]_{1,1} = \begin{cases} \kappa^{-1}\sigma^2 \rho_k e^{-\gamma\tau}\left(\cosh(\alpha_k\tau) - \frac{\gamma}{\alpha_k}\sinh(\alpha_k\tau)\right) & \gamma > \omega_k \\ \kappa^{-1}\sigma^2 \rho_k e^{-\gamma\tau}(1 - \gamma\tau) & \gamma = \omega_k \\ \kappa^{-1}\sigma^2 \rho_k e^{-\gamma\tau}\left(\cos(\alpha_k\tau) - \frac{\gamma}{\alpha_k}\sin(\alpha_k\tau)\right) & \gamma < \omega_k \end{cases}$$

Pulling it all together,

$$\mathbb{E}_{ss}\left[\|\Delta_t\|^2\right] = \frac{\eta^2 \sigma^2}{S(1-\beta^2)}\left(\mathrm{tr}(H)t + 2t\sum_{k=1}^{t}\left(1 - \frac{k}{t}\right)\sum_{l=1}^{m}\rho_l C_l(k)\right)$$

where $C_l(k)$ is defined as

$$C_l(k) = \begin{cases} e^{-\gamma k}\left(\cosh(\alpha_l k) - \frac{\gamma}{\alpha_l}\sinh(\alpha_l k)\right) & \gamma > \omega_l \\ e^{-\gamma k}(1 - \gamma k) & \gamma = \omega_l \\ e^{-\gamma k}\left(\cos(\alpha_l k) - \frac{\gamma}{\alpha_l}\sin(\alpha_l k)\right) & \gamma < \omega_l \end{cases}$$

for $\gamma = \frac{1-\beta}{\eta(1+\beta)}$, $\omega_l = \sqrt{\frac{2}{\eta(1+\beta)}(\rho_l + \lambda)}$, and $\alpha_l = \sqrt{|\gamma^2 - \omega_l^2|}$.

### G.3 TRAINING LOSS AND EQUIPARTITION THEOREM

In addition to solving for the expected values of the local and global displacements, we can consider the expected training loss and find an interesting relationship to the *equipartition theorem* from classical statistical mechanics.

The regularized training loss is $\mathcal{L}_\lambda(\theta) = \frac{1}{2}(\theta - \mu)^\intercal H(\theta - \mu) + \frac{\lambda}{2}\|\theta\|^2$, where $H$ is the Hessian matrix and $\mu$ is the true mean. Taking the expectation with respect to the stationary distribution,

$$\mathbb{E}_{ss}\left[\mathcal{L}_\lambda(\theta)\right] = \frac{1}{2}\mathrm{tr}\left((H + \lambda I)\mathbb{E}_{ss}\left[\theta\theta^\intercal\right]\right) - \mu^\intercal H \mathbb{E}_{ss}\left[\theta\right] + \frac{1}{2}\mu^\intercal H\mu$$

The first and second moments of the stationary solution are

$$\mathbb{E}_{ss}\left[\theta\right] = \mu \qquad \mathbb{E}_{ss}\left[\theta\theta^\intercal\right] = \frac{\eta}{S(1-\beta)}\frac{\sigma^2}{2}(H + \lambda I)^{-1}H + \mu\mu^\intercal$$

Plugging these expressions in and canceling terms we get

$$\mathbb{E}_{ss}\left[\mathcal{L}_\lambda(\theta)\right] = \frac{\eta}{4S(1-\beta)}\mathrm{tr}\left(\sigma^2 H\right) + \frac{\lambda}{2}\|\mu\|^2$$

Define the kinetic energy of the network as $\mathcal{K}(v) = \frac{1}{2}m\|v\|^2$, where $m = \frac{\eta}{2}(1 + \beta)$ is the per-parameter "mass" of the network according to our previously derived Langevin dynamics. At stationarity,

$$\mathbb{E}_{ss}\left[\mathcal{K}(v)\right] = \frac{\eta(1 + \beta)}{4}\mathrm{tr}\left(\mathbb{E}_{ss}\left[vv^\intercal\right]\right) = \frac{\eta}{4S(1-\beta)}\mathrm{tr}\left(\sigma^2 H\right)$$

where we used the fact that $\mathbb{E}_{ss}\left[vv^\intercal\right] = \frac{1}{S(1-\beta^2)}\sigma^2 H$. In otherwords, at stationarity,

$$\mathbb{E}_{ss}\left[\mathcal{L}_\lambda(\theta)\right] = \mathbb{E}_{ss}\left[\mathcal{K}(v)\right] + \frac{\lambda}{2}\|\mu\|^2.$$

This relationship between the expected potential and kinetic energy can be understood as a form of the equipartition theorem.

# H   EXPERIMENTAL DETAILS

## H.1   COMPUTING THE HESSIAN EIGENDECOMPOSITION

Computing the full Hessian of the loss with respect to the parameters is computationally intractable for large models. However, equipped with an autograd engine, we can compute Hessian-vector products. We use the subspace iteration on Hessian-vector products computed on a variety of datasets. For Cifar-10 we use the entire train dataset to compute the Hessian-vector products. For Imagenet, we use a subset 40,000 images sampled from the train dataset to keep the computation within reasonable limits.

For experiments on linear regression, the Hessian is independent of the model (it only depends on the data) and can be computed using any model checkpoint. For all other experiments, the Hessian eigenvectors were computed using the model at its initial pre-trained state.

## H.2   FIGURE 1

We resumed training for a variety of ImageNet pre-trained models from Torchvision [16] for 10 epochs, with the hyperparameters used at the end of training, shown in table 1.

We kept track of the norms of the local and global displacement, $\|\delta_k\|_2^2$ and $\|\Delta_k\|_2^2$, every 250 steps in the training process, to keep the length of the trajectories within reasonable limits. $\|\delta_k\|_2^2$ is visualized directly, along with its 15 step moving average. We then fit a power law of the form $\alpha k^c$ to the $\|\Delta_k\|_2^2$ trajectories for each model, using the last 2/3 of the saved trajectories. We visualize the $\|\Delta_k\|_2^2$ trajectories along with their fits on a log-log plot.

| Model | Dataset | Opt. | Epochs | Batch size $S$ | LR $\eta$ | Mom. $\beta$ | WD $\lambda$ |
|---|---|---|---|---|---|---|---|
| VGG-16 | ImageNet | SGDM | 10 | 256 | $10^{-5}$ | 0.9 | $5 \times 10^{-4}$ |
| VGG-11 w/BN | ImageNet | SGDM | 10 | 256 | $10^{-5}$ | 0.9 | $5 \times 10^{-4}$ |
| VGG-16 w/BN | ImageNet | SGDM | 10 | 256 | $10^{-5}$ | 0.9 | $5 \times 10^{-4}$ |
| ResNet-18 | ImageNet | SGDM | 10 | 256 | $10^{-4}$ | 0.9 | $10^{-4}$ |
| ResNet-34 | ImageNet | SGDM | 10 | 256 | $10^{-4}$ | 0.9 | $10^{-4}$ |

Table 1: **Figure 1 experiments training hyperparameters.**

## H.3   FIGURE 2

For this figure we trained a linear regression model on Cifar-10, using MSE loss on the one-hot encoded labels. The hyperparameters used during training are shown in table 2.

At every step in training the full set of model weights and velocities were stored. The top 30 eigenvectors of the hessian were computed as described in appendix H.1, using 10 subspace iterations.

The saved weight and velocity trajectories were then projected onto the top eigenvector of the hessian and were visualized in black. Using the initial weights and velocities, the red trajectories were computed according to equation (5).

| Model | Dataset | Opt. | Epochs | Batch size $S$ | LR $\eta$ | Mom. $\beta$ | WD $\lambda$ |
|---|---|---|---|---|---|---|---|
| Linear Regression | Cifar-10 | SGDM | 4 | 512 | $10^{-5}$ | 0.9 | 0 |
| Linear Regression | Cifar-10 | SGDM | 4 | 512 | $10^{-5}$ | 0.99 | 0 |

Table 2: **Figure 2 experiments training hyperparameters.**

## H.4   FIGURE 3

For this figure we constructed an arbitrary Ornstein-Uhlenbeck process with anisotropic noise which would help contrast the original and modified potentials. We sampled from a 2 dimensional OU process of the form

$$d \begin{bmatrix} x_1 \\ x_2 \end{bmatrix} = A \left( b - \begin{bmatrix} x_1 \\ x_2 \end{bmatrix} \right) dt + 10^{-4} \sqrt{D} dW_t.$$

where we set $b = [-0.1, 0.05]^\intercal$ and arbitrarily construct $A$ such that it's eigenvectors are aligned with $q_1 = [-1, 1]^\intercal$ and $q_2 = [1, 1]^\intercal$ and it's eigenvalues are 4 and 1 as follows:

$$D = \begin{pmatrix} 4 & 0 \\ 0 & 1 \end{pmatrix}, \quad V = \begin{pmatrix} -1 & 1 \\ 1 & 1 \end{pmatrix}, \quad A = V^{-1} D V$$

The background for the left panel was computed from the convex quadratic potential $\Phi(x) = \frac{1}{2} x^\intercal A x - b x$. The background for the right panel was computed from the modified quadratic $\Psi(x) = \frac{1}{2} x^\intercal U x - u x$, with $U = (D + Q)^{-1} A$ and $u = U A^{-1} b$ (see equation (25)). Both were sampled in a regular $40 \times 40$ grid in $[-0.1, 0.1] \times [-0.1, 0.1]$.

## H.5 FIGURES 4, 5, AND 6

Starting with the ImageNet pre-trained ResNet-18 from Torchvision [16], we resumed training for 5 epochs with the hyperparameters used at the end of training, shown in table3. The top 30 Hessian eigenvectors were computed as described in appendix H.1, using 10 subspace iterations. During training, we tracked the projection of the weights and velocities onto eigenvectors $q_1$ and $q_{30}$.

For Figure 3, we show the projection of the position trajectory onto eigenvectors $q_1$ and $q_{30}$ in 2D in black. The background for the left and center panels was computed taking the ImageNet pre-trained ResNet-18 from Torchvision [16] and perturbing its weights in the $q_1$ and $q_{30}$ directions in a region close to the projected trajectory. The training and test loss were computed for a grid of $20 \times 20$ perturbed models. The background for the right panel was computed according to equation (10).

| Model | Dataset | Opt. | Epochs | Batch size $S$ | LR $\eta$ | Mom. $\beta$ | WD $\lambda$ |
|---|---|---|---|---|---|---|---|
| ResNet-18 | ImageNet | SGDM | 5 | 256 | $10^{-4}$ | 0.9 | $10^{-4}$ |

Table 3: **Figures 4,5,6 experiments training hyperparameters.**

## H.6 FIGURE 7

We resumed training for an ImageNet pre-trained ResNet-18 from Torchvision [16] for 2 epochs, using the sweeps of hyperparameters shown in table 4. We indicate a sweep in a particular hyperparameter by $[A, B]$, which denotes 20 evenly spaced numbers between $A$ and $B$, inclusive.

We kept track of the norms of the local and global displacement, $\|\delta_k\|_2^2$ and $\|\Delta_k\|_2^2$, every step in the training process. The value for $\|\delta_k\|_2^2$ at the end of the two epochs is shown in the top row of the figure. We fitted power law of the form $\alpha k^c$ to the $\|\Delta_k\|_2^2$ trajectories for each model on the full trajectories. The fitted exponent $c$ for each model is plotted in the bottom row of the figure.

| Model | Dataset | Opt. | Epochs | Batch size $S$ | LR $\eta$ | Mom. $\beta$ | WD $\lambda$ |
|---|---|---|---|---|---|---|---|
| ResNet-18 | ImageNet | SGDM | 2 | [32, 1024] | $10^{-4}$ | 0.9 | $10^{-4}$ |
| ResNet-18 | ImageNet | SGDM | 2 | 256 | $[10^{-3}, 10^{-5}]$ | 0.9 | $10^{-4}$ |
| ResNet-18 | ImageNet | SGDM | 2 | 256 | $10^{-4}$ | [0.8, 0.99] | $10^{-4}$ |

Table 4: **Figure 7 experiments training hyperparameters.**

## H.7 INCREASING RATE OF ANOMALOUS DIFFUSION

Upon further experimentation with the fitting procedure for the rate of anomalous diffusion explained in Figure 1, we observed an interesting phenomenon. The fitted exponent $c$ for the power law relationship $\|\Delta_k\|_2^2 \propto k^c$ increases as a function of the length of the trajectory we fit to. As can

be seen in Figure 7, $c$ increases at a diminishing rate with the length of the trajectory. This could be indicative of $\|\Delta_k\|_2^2$ being governed by a sum of power laws where the leading term becomes dominant for longer trajectories.

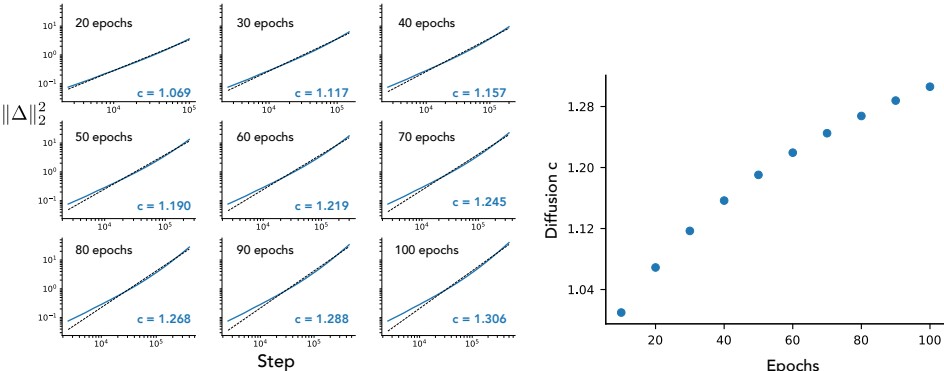

Figure 7: **The fitted rate of anomalous diffusion increases with the length of trajectory fitted.** The left panel shows the fitted power law on training trajectories of increasing length from the pre-trained ResNet-18 model. The right panel shows the fitted exponent $c$ as a function of the length of the trajectory.

