# OpenReview forum: "Rethinking the limiting dynamics of SGD: modified loss, phase space oscillations, and anomalous diffusion"
_ICLR.cc/2022/Conference — ICLR 2022 Submitted_

### Official Review · Reviewer_sDRt · 2021-10-29

**Correctness:** 3
**Technical Novelty And Significance:** 2
**Empirical Novelty And Significance:** 4
**Recommendation:** 5
**Confidence:** 5

**Main Review:**

In the current format, the paper suffers from several limitation that should be addressed by the authors. I believe that there are interesting results and I encourage the authors to answer my comments as I would be very happy to rise my recommendation score.
The following comments try to follow the section in the submission:




The authors in their analysis of related works miss a large portion of the relevant literature. Discussing the analysis of the dynamics of neural networks:
* The mean-field limit was introduced by three different independent works [Rotskoff, Vanden-Eijnden 2018] [Chizat, Bach 2018] [Song et al. 2018], only the ladder is cited.
* If the mean-field limit is mentioned then also the dynamical mean field theory approach should be cited. In the context of neural networks we have: [Mignacco et al. 2020][Mannelli et al. 2020][Mignacco et al. 2021].
* The dynamics of stochastic gradient descent and momentum were also obtained in dynamical mean field theory by the two papers by Mignacco above and [Mannelli, Urbani 2021].
* When mentioning the modelling of SGD using SDE, it is important to discuss the results of [Simsekli et al. 2019] where they showed that the jumps are fat-tail distributed and how that is compatible with the approximation.
* On the "empirical exploration" the authors refer only to [Papyan 2018], the results of [Ghorbani, Krishnan, Xiao 2019] that derived the same method independently and make thorough analysis of the dynamics.  In the same section also [Sagun et al. 2016] (ref [35]) should appear.

On page 2, the authors say "Surprisingly, [..] the networks continue to move through parameter space" but this very well known. It known that stochastic gradient descent keeps moving after training, indeed this is at the basis of ref. [16]. What is not known is how is moving, and indeed this finding is interesting.

In figure 1 lower panel, there is only one value on x-ax. Please add at least a second one. I can guess that the next number after 10^4 would be 2*10^4 but this is not obvious a priori.

The authors should try to avoid to add adjectives to embellish the paper in the technical sections. These parts of the paper should be factual and reporting their results in a scientific way.
For instance in section 8, describing figure 5 the authors say "As predicted, the spiral appears more evident ..", can you quantify how "spiral" the plot is? It is hard to conclude from that picture. Furthermore, that appears to be the result of one run. It should be good to average over many simulations to avoid the risk of observing random fluctuations.
The fact that the second figure appears "less evident" may indicate a limit in the theory. Since the authors are considering a SDE, higher order effects should appear more frequently for eigenvectors corresponding to smaller eigenvalues.
It would be interesting to see what happens to the 1000th eigenvectors. Since ImageNet has 1000 classes, according to [Sagun et al. 2016][Papyan 2019] that eigenvector should be associated to irrelevant information and probably this behavior disappear. You could observe this also if you have the pretrained network in CIFAR100 or CIFAR10, by looking at the 100th and 10th eigenvector respectively.

An important aspect, that has not been discuss, is the effect on generalization. What are the practical consequences of these results? Although the theoretical framework can not be applied to the test dataset, the authors can verify the practical effects.
Is being super/sub diffusive an advantage? Does the performance change? Answering these questions will bring value to the work.

The authors affirm that the dynamics brake detailed balance. Unfortunately I must have missed this part in the paper. Could they clarify?
Also, what are the practical implications of braking detailed balance?

In figure 6 the authors claim that their theory predicts the behavior of the displacement. It is not clear to me the procedure used. When you say "using this single estimate" what do you mean? Are evaluating \sigma^2 tr(H) in a single point for each plot and using this information in draw the line? It is not very clear.
On the second line, it is an overstatement to say that you predict the diffusion constant. Your model, by construction, is a correlated Brownian  motion with a drift, therefore the diffusion constant is 1.
This is a pity, there is no explanation for the very interesting behavior the you observe.



**Summary Of The Paper:**

The paper uses a stochastic differential equation approximation of stochastic gradient descent momentum to model the neural networks dynamics. At the theoretical level the study focuses on linear regression. They showed that there is agreement between the results in the simpler model and deep neural networks.
The most relevant fact is that the networks can have a super or sub diffusive behavior at the end of training. Unfortunately, their theory cannot explain this finding nor the practical consequences on the test loss.

**Summary Of The Review:**

The paper presents an interesting result about the presence of sub/super diffusive behaviour at the end of training. Although the result may be of limited practical interest, it is a new finding and would be interesting to understand causes and consequences of that (not discussed in the paper).
The paper has a large number of flaws that I believe can be amended to get a publishable result.

---

> ### Author Response · Authors · 2021-11-10
> **Thank you for your review**
>
> Thank you for your review and we are glad to hear that our paper “presents an interesting result about the presence of sub/super diffusive behaviour at the end of training.”
>
> Let us address and clarify your main points here:
>  - **Related works:** we will be sure to add the citations to [Rotskoff, Vanden-Eijnden 2018], [Chizat, Bach 2018], [Mignacco et al. 2020], [Mannelli et al. 2020], [Mignacco et al. 2021], [Mannelli, Urbani 2021], [Simsekli et al. 2019], [Ghorbani, Krishnan, Xiao 2019].
>  - **Surprisingly:** We will remove the word “Surprisingly” from page 2.
>  - **Fig. 1:** We will edit the x-axis of figure 1 as suggested.
>  - **Adjectives:** We will remove adjectives and adverbs that embellish the paper in section 7 and 8.
>  - **Qualitative versus Quantitative:** We separate our empirics into a qualitative section (section 7) and a quantitative (section 8).  So all the results in section 7, including the “spiral” plot (fig. 6) are qualitative by nature and there is not actually a good way to quantify these results.  That is what section 8 does.
>  - **Generalization:** Directly constructing a theory of deep learning explaining generalization performance is extremely difficult.  In this work, we take a phenomenological route that has proven to be promising to unlocking mysteries in deep learning (for example Double Descent (Belkin et al., 2018) or Spherical Motion Dynamics (Wan et al., 2020)).  By identifying phenomena, explaining them in a simpler setting, and providing evidence of these predictions at scale we are furthering our understanding of deep learning.  But, we agree that there is clearly much more work to elucidate a connection to generalization and are certainly thinking in that direction as well.
>  - **Broken detailed balance:** As we discussed in section 6, broken detailed balance occurs when the probability currents are non-zero.  So non-zero probability currents or broken detailed balance are the same thing.  The consequence of broken detailed balance in phase space is what causes the anomalous diffusion!
>  - **Figure 6:** We estimate $\sigma^2 \mathrm{tr}(H)$ by training a single model with the default hyperpapermaters and inverting equation 11 (this is the black dot in the figure).  Using this single estimate we can then predict over a large sweep of different hyperparameters (the black dotted line).  We then train 60 independent experiments with unique hyperparameter combinations and plot their empirical values as the colored circles in the figure.
>  - **Exponent of anomalous diffusion:** We did not claim to predict the exponent of anomalous diffusion, which is why we write “Unfortunately, it is not immediately clear how to extract the explicit exponent $c$ from equation (12)”, rather our goal was to “predict overall trends in the diffusion exponent $c$.”  We changed the title of this section from “Predicting the diffusive behaviour...” to “Understanding the diffusive behaviour…”  We hope this will help avoid confusion.  Are there other changes you would like to see in order to make this more clear?
> - **Diffusion constant is 1:** If there was no temporal correlation in the noise then you would be correct the exponent of diffusion would be 1.  But there is temporal correlation, which is why we observe anomalous diffusion.  We discuss this in detail in section 8 under “Exponent of anomalous diffusion”.
>
> We thank you again for your review and detailed comments and questions.  But to be frank given your feedback we do not understand why you gave us a review of 3/10.  We hope you would consider raising your score to accept.  If you do not agree we hope to have a discussion so we can understand your reservations and hopefully clarify our work.  Thank you!

---

> > ### Comment · Reviewer_sDRt · 2021-11-12
> > **Thank you for your reply**
> >
> > Thank you for the clarifications.
> >
> > I am happy that you are modifying the manuscript in line with my comment, but I could not find the updated version of the paper yet.
> >
> > *Qualitative vs quantitative:* Maybe the new version of the paper is not online yet, but the spiral plot (which is in fig.5 in the version that I can see) says in the caption " As predicted [...]". In the comment I was complaining about the fact that: 1. this is more an observation that a prediction, 2. I honestly can not be sure of the statement just looking at the plot of a curve in 2D. You should include a measurement of the speed of rotation. I am not claiming that you must provide a quantitative explanation, but simply that you should show the reader quantities that he can recognize in what is stated without having to stare at the figure to see if that may or may not be true.
> >
> > *Generalization:* I am aware of the approach that you are using, and I believe it is a good direction to tackle open problems in the field. My concerns is specific to this submission and not the entire idea behind, and in particular I am struggling with find its significance. The paper shows some behavior at the end of training and shows that there are analogies with OU processes, but then does not use the analogy to build understanding. The question of what your observation implies: both in the training and test errors are missing. I do not understand why (for instance) changing the learning rate leads to anomalous diffusion or why the dynamics, without changing parameters, seems to converge to a stationary process with diffusion constant 1.
> >
> > Reading the other reviews, I realized that the idea of anomalous diffusion was already in the paper by [Baity-Jesi et al. 2018] (1803.06969). So I am looking forward to your reply to reviewer "pbf9".
> >
> > I would like to clarify the reason behind my score. I see this work as a minor contribution to the field that is marginally below the level of acceptance for ICLR. Then I lowered my score according to the flaws that believe are in the paper.

---

> > > ### Author Response · Authors · 2021-11-12
> > > **Thank you for your reply**
> > >
> > > Thank you for your reply. Sorry for the delay, the updated manuscript will be uploaded this weekend. We will follow up once we post it.  In the meantime here is a reply to your comments:
> > >
> > > **Fig.5:** The caption will be changed.  You are correct it should not say “As predicted [...]" and we agree this figure is not the most compelling qualitative evidence (fig. 4 and 5 are better for that).  We can highlight this point as well in the caption.
> > >
> > > **Generalization:** Here is a snippet from our reply to Reviewer 4wwS explaining why the study of limiting dynamics is important to generalization:
> > > > Consider [Izmailov et al., 2018], one of the papers we mentioned in our list.  The authors of this work found that “that simple averaging of multiple points along the trajectory of SGD, with a cyclical or constant learning rate, leads to better generalization than conventional training”.  This work studies pretrained models (in fact the exact same default Pytorch models we studied) and finds “SWA [Stochastic Weight Averaging] provides consistent improvement by 0.6-0.9% over the pretrained models.”  Follow up works on SWA, also leveraging the limiting dynamics of pretrained neural networks, include [Maddox et al. 2019] and [Izmailov et al., 2020].  Prior works to SWA, also studying pretrained neural networks, include [Garipov et al. 2018], who found that Fast Geometric Ensembling (FGE) of a pretrained neural network could improve the final performance (this is the connection to ensembling).  As explained by Garipov et al., “We used a pretrained model with top-1 test error of 23.87 to initialize the FGE procedure. We then ran FGE for 5 epochs....the top-1 test error-rate of the final ensemble was 23.31. Thus, in just 5 epochs we could improve the accuracy of the model by 0.56 using FGE”. All these works demonstrate that even “after” training state-of-the-art neural networks, there is still lots of room for performance improvement.  Thus, the theoretical and empirical study of limiting dynamics is **not** trivial and has yielded useful results.
> > >
> > > **Building understanding:** It's hard to understand why you think our work “does not use the analogy to build understanding”?  As reviewer pbf9 pointed out, they “appreciate the clarity of the assumptions that are needed to obtain every single result, while maintaining the discussion at an intuitive level.”  A central goal of this work was not to just leave equations on a page, but actually walk the reader through the derivation and intuition to build understanding.  Understanding such as, in section 5 we explain how in the setting of linear regression, we can use the OU analogy of SGD to “decompose the expectation as a sum of harmonic oscillators in the eigenbasis {q1, . . . , qm} of the Hessian”, which we verify empirically in figure 2.  In section 7, we describe how we can use the equation for the modified loss to understand that at the end of training the “training trajectory of the network will behave isotropically, since it is driven not by the original anisotropic loss, but a modified isotropic loss”, which again we verify qualitatively in figure 3. And finally in section 8 we use the results of the stationary distribution to understand quantitatively the constant instantaneous speed and exponent of anomalous diffusion empirics observed in section 2.  You seem to be fundamentally misunderstanding how a stationary Gaussian process in broken detailed balance could lead to anomalous diffusion.  See the follow up post below for further explanation.
> > >
> > > **[Baity-Jesi et al. 2018]:** We were also unaware of this work until reviewer pbf9’s review.  But please read through the paper and ask yourself whether “the idea of anomalous diffusion was already in the paper”.  The main conclusions of [Baity-Jesi et al. 2018] are that there are three stages of training which can be observed both in the dynamics of the loss and mean squared displacement. Yet, **nowhere in this work  will you find the term “anomalous diffusion”** or a careful description of the exponent of the power law relationship in the final stage of training.  Indeed, even this work is not the first to study “Euclidean distance of weight vector“ in deep learning  such as found in Hoffer et al, 2017, whom we currently discuss in section 2. Moreover, the mechanisms Baity-Jesi et al. invoke are fundamentally different: they are inspired by glassy systems. In contrast, we analytically derive phase space oscillations in a quadratic bowl at the end of training with incommensurate frequencies.  We will discuss and cite this work in section 2 in our updated manuscript.
> > >
> > > Thank you again for your review and we hope you will consider raising your score to accept.

---

> > > > ### Author Response · Authors · 2021-11-12
> > > > **Understanding Anomalous Diffusion**
> > > >
> > > > You state in your original review “your model, by construction, is a correlated Brownian motion with a drift, therefore the diffusion constant is 1”.  This doesn’t make sense. For example, an OU process in one dimension at statiority will not demonstrate diffusion at all (let alone with exponent 1), as by definition of stationarity the drift balances the diffusion.  So the question “how can a high-dimensional gaussian process with stationary distribution exhibit anomalous diffusion?” is very good and is a central message of our paper .
> > > >
> > > > It is not initially obvious, so let's start simpler and consider a particle with coordinates $x_1, \dots, x_N$ diffusing isotropically in an isotropic  quadratic well in $N$ dimensional space.  The stationary distribution of this particle is a Gaussian with identity covariance in $N$ dimensional space.  In high dimensions, such a distribution concentrates on the surface of a sphere of radius $\sqrt{N}$ (i.e. $x_1^2 + \dots + x_N^2$ concentrates on the value $N$).  And diffusion in this high dimensional space in the quadratic well can be thought of as a random walk on the high dimensional sphere, which has surface area exponential in $N$.  If one starts on this sphere and does such a random walk, one is highly unlikely to return close to where one started, unless one waited an amount of time that is exponential in N (due to the exponentially large surface area).  Thus, for all experiments where one tracks the distance of the particle from its initial position for times less than exponential in N, one is overwhelmingly likely to see unbounded diffusive growth of the distance, growing as $\sqrt{t}$.  This is a well understood result in high dimensional random diffusion in a quadratic well or very similarly, random walks on a high dimensional sphere.  One simply cannot detect the confining nature of the quadratic well, or the finite surface area of the sphere through the growth of distance travelled, unless one waits an amount of time exponential in dimension.  Thus, in contradiction to the one dimensional setting, due to high dimensionality,  a Gaussian process with stationary distribution can indeed exhibit global displacement growing monotonically for very long times, and in this case because of the isotropy of the original well, the exponent of growth would be $c =1$.
> > > >
> > > > A similar argument holds if each dimension is oscillating, with frequencies/periods of oscillation that are incommensurate (have no common divisor) which leads to anomalous diffusion $c \neq 1$.  To understand this better consider a trajectory in $2$ dimensional space with coordinates $x_1$ and $x_2$.  Suppose $x_1$ oscillates with a period of $2$ seconds and $x_2$ oscillates at a period of $4$ seconds. Then because the faster oscillator with period $2$ has a period that divides that of the slower oscillator,  every $4$ seconds the combined system of two oscillators will return to its original point in the $2$ dimensional space $(x_1,x_2)$.  However, if the two periods are incommensurate, this will not be the case, and the dynamics will not return to the original point and instead fill up space and at a rate controlled by the regime of the oscillation (i.e. underdamped, critically damped, overdamped).  In extremely high dimensions, in which you have $N$ oscillators with incommensurate frequencies, the combined $N$ dimensional system will generically also not return to its original state, nor even come close to it, in a time that is less than exponential in $N$.   In this fashion, many oscillators with incommensurate frequencies together exhibit growing global displacement even if individual oscillators always return to their original state.  And in contrast to the isotropic setting previously, the different regimes of oscillation will lead to different regimes of diffusion (i.e. sub-diffusion, standard diffusion, super-diffusion). In deep learning the eigenvalues of the Hessian are generally different real numbers, thus our oscillator system describing the limiting dynamics will exhibits anomalous diffusion.  And as explained in section 8 the hyperparameters of optimization (learning rate included) will control the regime of the oscillators and in turn the exponent of anomalous diffusion.
> > > >
> > > >
> > > > Overall, we hope this understanding of diffusion and oscillations in high dimensions addresses your concerns that our theory does not build an understanding of our observed phenomena.  If you think it would be helpful for the reader we can add this explanation to the appendix of the paper for reference.

---

> > > > ### Comment · Reviewer_pbf9 · 2021-11-14
> > > > **Since I'm mentioned, I will add something**
> > > >
> > > >
> > > >
> > > > Anomalous diffusion in DNNs is not something new. The authors claim that their result is novel because the exact wording was not used in arXiv:1803.06969. However, it was also measured e.g. in arxiv:2009.10588, where the term **anomalous diffusion appears in the title**. As for the explanation of the underlying phenomenon, I do not see how this is relevant, since here we are simply referring to the empirical observation of a $c\neq1$ exponent.
> > > >
> > > > I also second reviewer sDRt's comment on giving a quantitative measure of the spiral motion.

---

### Official Review · Reviewer_pbf9 · 2021-11-01

**Correctness:** 3
**Technical Novelty And Significance:** 2
**Empirical Novelty And Significance:** 2
**Recommendation:** 6
**Confidence:** 4

**Main Review:**

The paper is well written; a very pleasant reading. I find interesting the characterization of diffusion and displacement as a function of hyperparameters. Although it is not new that SGD obeys a modified loss, I find it insightful to see that this modified loss can be isotropic despite the loss being anisotropy, and to learn under which conditions this occurs. I also appreciate the clarity of the assumptions that are needed to obtain every single result, while maintaining the discussion at an intuitive level.


Major Criticisms.

- Result (a) is not new. The anomalous diffusion at the end of training was already shown in arXiv:1803.06969, exactly in the terms defined in this manuscript ("distance travelled grows as a power law in the number of gradient updates with a nontrivial exponent").

- Result (c). That SGD obeys a modified loss was also stated in previous literature, such as Ref.[21] or, in a different framework, arxiv:1803.01927.

- The title is very bold, inviting a rethinking of the limiting dynamics. I would say that this is because the authors assume that the deep learning community thinks that [statement (g) up here] “the network converges in parameter space”, or “the network stays within a local region”. These are however very naive intuitions that nobody with sufficient experience in deep learning theory would believe. Also, one of their main claims is [statement (f)] that detailed balance does not hold, implicitly suggesting that the results of e.g. Refs.[16,23] could be flawed. Again, nobody believes that detailed balance holds for SGD dynamics -not even at late stages of training-, since detailed balance is a sufficient condition (not even necessary, see e.g. https://aip.scitation.org/doi/full/10.1063/1.4863991) for equiibrium, but steady state distributions generally do not satisfy it. There is a whole section in Ref.[16] devoted to showing that SGD is out of equilibrium. Another example is arXiv:1803.06969, where they state that at the end of learning the system diffuses at the bottom of the landscape with a time-dependent diffusion constant (so this is clearly not an equilibrium process, and the network does not stay confined within a local region). In other words, I don't think that statements (f) and (g) add anything new to the current knowledge.

- The analysis of the limiting cycles is nice, but I do not see what we learn more than was already presented in Ref.[21]. The only novelty would be that the paths at the end of learning are not limit cycles, but rather space filling curves. However, in the way as it is presented, this looks more like a conjecture than an actual result. As for the description in terms of velocities, this is nice, but I don't find it surprising that the velocities oscillate, especially in the directions of the top 30 eigenvectors. A system confined by quadratic walls will have an oscillating velocity, as any harmonic oscillator. I would find it more surprising if the authors measured the same kind of oscillation in the direction of the bottom eigenvectors.


Minor comments:

- The authors project the limiting trajectory for the parameters onto the plane spanned by the top q 1 and bottom q 30 eigenvectors. I assume that  q1 and q30 were chosen in order to maximize the anisotropy on the related plane, in order to show that the trajectory is instead isotropic. I suggest to explain that this is the reason of the choice, since at first it wasn't clear to me why this was being done.

- Fig.3. It's nice to see that the trajectory is isotropic when no regularization is used. But why is the trajectory not centered around the minimum of the modified loss? Also, it looks like the minimum of test and modified losses coincide. Why is this?

- typo in section 8: trigonomentric

- Implicit regularization of the velocity trajectory. I assume that the authors mean that the velocity appears in the modified loss as an L2 regularization term. Is this right? Is there any further consequence to this?

- In figure 6 - bottom, the dots are measured from fits of the trajectory. The dashed line is c=1 diffusion, but this is a little confusing, because the dashed line in the top three plots is the theoretical prediction of Eq11.

**Summary Of The Paper:**

The paper studies the long-time dynamics of deep neural networks. The author(s) (1) show some empirical findings related to the mean square displacement, (2) model SGD as an underdamped Langevin Equation, relate it to an Ornstein Uhlenbeck process in a linear regression setting, and use it to study the limiting dynamics of SGD, (3) use the Fokker-Planck formalism to show that the steady state weight distribution obeys a modified loss which is isotropic in the absence of L2-like regularization, (4) provide empirical evidence that their findings are relevant also beyond the context of linear regression.
According to the authors, these are the novelties provided by the paper:

(a) "We find empirically that long after performance has converged, networks continue to move through parameter space by a process of anomalous diffusion in which distance travelled grows as a power
law in the number of gradient updates with a nontrivial exponent."

(b) "We reveal an intricate interaction between the hyperparameters of optimization, the structure in the gradient noise, and the Hessian matrix at the end of training that explains this anomalous diffusion."

(c) "we show that the key ingredient driving these dynamics is not the original training loss, but rather the combination of a modified loss, which implicitly regularizes the velocity, and probability currents, which cause oscillations in phase space"

(d) "We identify qualitative and quantitative predictions of this theory in the dynamics of a ResNet-18 model trained on ImageNet."

(e) "We uncover a mechanistic origin for the anomalous limiting dynamics of deep neural networks trained with SGD"

(f) "one of the most significant results of our analysis is that, depending on the relationship between the gradient noise covariance and the Hessian, the stationary distribution in phase space will generically violate detailed balance"

(g) 'Natural intuitions, such as “the network converges in parameter space” or “the network stays within a local region”, are wrong'


**Summary Of The Review:**

The paper presents several concepts in a very clear and readable manner. However, the novelties presented are limited/incremental (certainly not as revolutionary as suggested by the title), and some of the results are either not new (anomalous diffusion in DNNs has been known for years now), not relevant (the lack of detailed balance in results relying on a stationary state distribution), or not well-supported (space filling curves instead of limit cycles). On the other hand, I do find some value in some of the results, such as the characterization of diffusion and displacement as a function of hyperparameters, and showing that under certain conditions the modified loss is isotropic.

The authors should reduce their claims regarding what is new in their work and adapt them to what is the real state of the art of the understanding of DNNs, and remove disputable statements such as "The major limitation of the stationary dynamics approach to analyzing SGD is its implicit assumption that the system is in detailed balance".

---

> ### Author Response · Authors · 2021-11-12
> **Thank you for your review**
>
> Thank you for your review and we are glad to hear that you find our work “well written”, “interesting”, and “insightful”.
> - **Result (a):** You are correct that the work by [Baity-Jesi et al. 2018] studies the dynamics of the global displacement through different phases of training on a variety of small architectures and datasets. We were not aware of their work and we will cite them in section 2. That said, it seems like quite a misrepresentation of their work and ours to claim that there is significant overlap, even in the empirics. We have not studied this work, but it seems that the main conclusions are that there are three stages of training which can be observed both in the dynamics of the loss and mean squared displacement (please correct us if we are wrong). Nowhere in this work  will you find the term “anomalous diffusion” or a careful description of the exponent of the power law relationship in the final stage of training.  Indeed, even this work is not the first to study “Euclidean distance of weight vector“ in deep learning  such as found in Hoffer et al, 2017, whom we currently discuss in section 2. Moreover, the mechanisms Baity-Jesi et al. invoke are fundamentally different: they are inspired by glassy systems. In contrast, we analytically derive phase space oscillations in a quadratic bowl at the end of training with incommensurate frequencies. Indeed this understanding for how anomalous diffusion can arise from a stationary Gaussian process in broken detailed balance in high dimensions is novel. Please see our reply to reviewer sDRt for more details.
> - **Result (c) is not new:** We never claimed to be the first to say “SGD obeys a modified loss”. Indeed we cite many works that have made similar arguments.  You misrepresented our work in your summary when you wrote “according to the authors, these are the novelties provided by the paper: (a) - (g). We clearly define three novel contributions in the “significance” and “originality” section on page 9.
> - **The title is very bold:** We assume your concern is with the word “Rethinking” (as the rest of the title is essentially an extremely shortened version of our abstract)? You are correct about our motivation for using the word rethinking, but we are not tied to it and if you think this word could be improved or should be removed to make this work clearer, we would be happy to consider it.
> - **Detailed balance:** We will remove the sentence “The major limitation of the stationary dynamics approach…”. We agree with you this was too strong and it was not our intent to “implicitly suggesting that the results of e.g. Refs.[16,23] could be flawed”.
> - **Originality with respect to [Chaudarhi & Soatto, 2018]:** The work by Chaudarhi & Soatto proposed the existence of a modified loss and probability currents driving the limiting dynamics of neural networks, but did not derive explicit expressions. In our work we extend their analysis from weight space to phase space, incorporate the effect of momentum, and by restricting ourselves to the setting of linear regression, we are able to derive explicit expressions (equation 8 in our work). Additionally, given our setting we were able to remove the opaque assumption that the “force $j(x)$ is conservative” (assumption 4 in their work). Because we derived explicit expressions, we could demonstrate empirically predictions of this theory for deep networks trained on large scale datasets. Further, these expressions actually demonstrate that one of the main claims made by Chaudarhi & Soatto, namely that SGD converges to limit cycles for deep networks (as clearly stated in their title), is actually not the full picture. By explicitly identifying the stationary probability current as oscillations characterized by the eigenvalues of the Hessian, then the only way these currents are limit cycles is if all non-trivial eigenvalues of the Hessian are integer multiples of each other. For real-world datasets, with a large spectrum of incommensurate frequencies, we should expect the trajectories following probability currents to be space-filling curves rather than limit cycles! Our theory and empirics are a significant contribution to validating and deepening our understanding of the concepts originating in Chaudarhi & Soatto.
>
> Thank you again for your detailed review. We found it very helpful and have implemented all of your suggestions (will update manuscript shortly). We see our work as an important and timely addition to the literature demonstrating empirically/theoretically clear effects of stochasticity, highlighting the analysis of dynamics in phase space, clarifying misconceptions in the field around the presence of limit cycles, and demonstrating how a system of incommensurate oscillators can result in anomalous diffusion. We hope you would consider raising your score to accept. If you still have reservations we would be happy to continue a discussion hopefully clarifying our work and contributions. Thank you!

---

> > ### Comment · Reviewer_pbf9 · 2021-11-14
> > **Reply**
> >
> > (a) I fail to see how my comment is a misrepresentation of either the authors' work, or arXiv:1803.06969, since my comment refers to empirical observations and not to the underlying mechanisms. That paper measures a mean square displacement with an exponent different from 1, so they are measuring anomalous diffusion. I do not see how the explicit use of the word is important (nor, obviously, had I checked whether the term is used). However, this does not change anything, because while that is (to my knowledge) the first paper measuring such a phenomenon, anomalous diffusion was also measured later on, e.g. in arxiv:2009.10588, where the term "anomalous diffusion" is in the title.
> >
> > (c) The points (a)-(g) are verbatim quotations from the manuscript. As the authors say, they are not solely taken from the final sections of the paper. To the contrary, I carefully read the manuscript, collected the claims throughout it, and pasted them into the report. Since these claims are exactly copied from the manuscript, it is hard to underrstand why the authors attack this list. An explanation is that despite the manuscript being easily readable, there is a problem with how it is written, in that some times old results can be confused as claims of novelty. This point was raised already by other reviewers. As for (c) in particular, it is hard for me to assess what the authors mean. A constructive reply by the authors would have allowed me to address this point.
> >
> > Title. Yes, I refer to the word "rethinking", and I would appreciate it being removed.
> >
> > Detailed balance. Ok.
> >
> > Limit cycles. I see the point. Wouldn't it be possible to make the same argument (space filling curves) simply from the results at Ref 21? This does not mean that it doesn't make sense to write it, but would the authors mind clarifying why their construction allows to argue in favor of space filling curves? Also, what changes in practice if the scenario is one or the other?
> >
> > Finally, I find that the tone towards several comments by more than one reviewers is inappropriate. I would invite the authors to use a more constructive attitude.

---

> > > ### Author Response · Authors · 2021-11-15
> > > **Thank you for your reply**
> > >
> > > A: We were not aware of arxiv.org/abs/2009.10588, which we clearly should have been.  We will rewrite section 2 and the abstract and conclusion to clearly explain how we are not the first to identify the observations of anomalous diffusion.  We will discuss arXiv:1803.06969 as well, as you explained.
> > >
> > > C: Maybe there has been a misunderstanding. Our intention was not to attack this list as a good overview of our work (which it is), but rather to clarify that we believe this list does not characterize the “novelties” of the paper  “according to the authors”, as stated in your summary.  Because of this, we don't understand why one of your major criticisms is that previous literature also identifies  a modified loss, as we never claimed to say otherwise.  Perhaps we are misunderstanding your original point?
> > >
> > > Limit cycles:  [21] directly states in Lemma 7 that the “most likely trajectories of SGD traverse closed trajectories in weight space”.  That said, I think this statement is likely too strong.  It is not clear why the probability currents identified by [21] are necessarily limit cycles, rather than being space filling curves, or at least, being space-filling in some subspace.   It is clear the curves are "rotational", due to the skew-symmetry, but it isn’t clear that anything prevents the probability current from (for example) wrapping around a submanifold with irrational period and thus being ergodic in that submanifold without ever coming back on itself exactly.  By studying the OU setting we were able to derive closed form expressions for the probability currents and therefore determine exactly what condition would be needed for the probability currents to be limit cycles or space filling curves.  This is what allowed us to conclude the latter setting is much more likely in a realistic learning setting.  In terms of what changes in practice, it is the space filling property which leads to the anomalous diffusion.  If the probability currents were actually limit cycles (even in high dimensions) then eventually the model would return to where it started.
> > >
> > > Your earlier minor comments and questions:
> > > - You are correct, we will add an explanation of this design choice.
> > > - We assume that the center of the modified loss is off the center for the trajectory because of inaccuracy in our estimate for the mean $\mu = \theta_* - H^{-1}_* g_*$.  Why the modified loss and test loss centers seem to overlap is an interesting observation.
> > > - Fixed typo, thanks.
> > > - You are correct.  We also wanted to highlight the similarity and difference with the recently published “Implicit Gradient Regularization”.
> > > - We will remove the line or reconsider labeling to make this more clear.
> > >
> > > Tone: We apologize if our tone has come off wrong, this was not our intention.  We greatly appreciate all the time you and the other reviewers have clearly spent reading, understanding, and discussing our work.  Our paper will be better off because of it.

---

> > > > ### Comment · Reviewer_pbf9 · 2021-11-17
> > > > **Thanks for addressing my points**
> > > >
> > > > I appreciate the authors taking into account my comments.
> > > > I will reply point by point:
> > > >
> > > > A. I look forward to the new version.
> > > >
> > > > C. The incriminated sentence is
> > > >
> > > > "we show that the key ingredient driving these dynamics is not the original training loss, but rather the combination of a modified loss, which implicitly regularizes the velocity, and probability currents, which cause oscillations in phase space"
> > > >
> > > > I think that part of the misunderstanding is because I misinterpreted this sentence as a claim regarding the modified loss. Instead, the authors are referring to the combination of modified loss and probability currents. However, I think that one can argue that this also is not new. I suggest that either the authors rephrase the sentence, or they provide a good argument for this claim.
> > > >
> > > > Limit cycles. Thanks for the explanation.
> > > >
> > > > I appreciate the mention to the minor comments and the change of tone.

---

### Official Review · Reviewer_4wwS · 2021-11-01

**Correctness:** 3
**Technical Novelty And Significance:** 2
**Empirical Novelty And Significance:** 2
**Recommendation:** 3
**Confidence:** 5

**Main Review:**

I welcome the general direction of the work: trying to understand the dynamics
of neural networks is a complex undertaking that a large community of
researchers is pursuing, so the study of simplified models is a promising
avenue.

### Setup of the study

In studying the *limiting* dynamics however, the authors limit themselves to a
setup where I don't see any immediate connections to learning or representations
of neural networks. The dynamics the authors describe happen *after* resuming
training of a pre-trained neural network, thus I feel like their setup restricts
the potential impact of the results of this study.

### Clarity / novelty of the results

I found the article hard to read at times because the authors repeatedly qualify
their observations as "surprising", "contrary to common intuition",
"nonintuitive", etc. I think these qualifiers can be mistaken as claims of
novelty etc. and would hence use them more sparingly. For example, the fact that
neural networks continue to move through their weight space has been
well-established for quite some time now, cf. for example [Jastrzebski et
al. '17, Chaudhari & Soatto, 18, Baity-Jesi et al. '18, and many more]. Hence I
didn't find the observation in Figure 1 "surprising" (p 2 after the equation),
which underlines the subjectiveness of these claims - or else I maybe reading
the figure incorrectly?

Another example: I would consider it well-established that OU processes whose
diffusion matrix is not isotropic do not follow the naïve Gibbs distribution,
but instead equilibrate in a modified potential (see for example Section 5.3
"Potential conditions" of Gardiner's "Handbook of stochastic methods" etc.)
Furthermore, modified losses arising through SGD dynamics have been studied in a
number of recent deep learning papers, some of which are cited by the authors

Other claims about the significance of the results should equally be clarified
in my opinion, for example:

> The expectation that the training trajectory would reflect the underlying
> anisotropy of the training loss driving the dynamics is also wrong;" (p. 9)

In my understanding, this study is only concerned with the *limiting* dynamics
of learning, and hence conclusions about the training cannot be drawn
immediately?

### Separation of experimental from theoretical results

The authors should be lauded for trying to connect their theoretical results
with empirical results on deep networks. Again though, I think the presentation
of the results should be revised to clarify which predictions are actually
derived from theory.  Take the exponent of anomalous diffusion (bottom of Fig
6): it cannot be estimated of the global displacement (12), as the authors
explain. Instead, the authors evaluate the dependence of the diffusion constant
on learning rate, batch size and momentum parameter directly from a simulation
by fitting a power-law to the empirical displacement. I would present this
result separately, as it is not a theoretical prediction, and thus presenting it
in a section entitled "Predicting the diffusive behaviour of the limiting
dynamics" could cause confusion in my opinion.


**Summary Of The Paper:**

The starting point of the present paper is the observation that the *parameters*
of a neural network continue to change when training the network with stochastic
gradient descent (SGD) even after the performance of the network has stabilised
at a final value. The authors study these "limiting dynamics" in linear
regression, which they model as an underdamped Langevin equation resulting in an
Ornstein-Uhlenbeck (OU) process and find an oscillatory dynamics in the
(position, velocity)-phase space. The authors discuss how to connect their
findings in this simple model with the dynamics of Resnet18 (see
below). Finally, they attempt to predict features of the diffusive behaviour of
the ResNet18 from their model (Sec. 8)

**Summary Of The Review:**

I welcome the direction of the present study, trying to understand the dynamics
of deep neural networks via simpler models. However, I cannot recommend
acceptance of the present paper for three reasons: the implications of the
studied dynamics on learning or representations in neural network remain unclear
to me; the results derived from the Ornstein-Uhlenbeck process do not go beyond
the previous work to an extent that would warrant the publication at ICLR in my
opinion; and the unclear presentation of the results which doesn't always
clearly separate previous work from the presented results, or theoretical
predictions from numerical measurements (Fig. 6).

**Edit Nov 14th** I thank the authors for the lengthy replies to the points raised in this review. After careful consideration of their comments, and also taking into account the valuable points made by other referees, I keep my original score for now. I have increased my confidence from 4 to 5.

---

> ### Author Response · Authors · 2021-11-10
> **Thank you for your review**
>
> Thank you for your review and we are glad to hear that our paper “should be lauded for trying to connect their theoretical results with empirical results on deep networks.”
>
> Let us address your three reasons for not accepting this paper:
>
> (1) **Why we should study simple theoretical settings or the limiting dynamics of deep neural networks.**
> There have been many works that have used theoretical and empirical insights of the limiting dynamics to construct hyperparameter schedules and algorithms to improve performance. Most famously, the *linear scaling rule*, derived by [Krizhevsky, 2014] and  [Goyal et al. 2017]  from a similar SDE-based analysis, relates the influence of the batch size and the learning rate, facilitating stable training with increased parallelism. This relationship was extended by [Smith et al. 2017] to account for the effect of momentum. [Lucas et al. 2018] proposed a variant of SGD with multiple velocity buffers to increase stability in the presence of a nonuniform Hessian spectrum, again derived from an analysis similar in spirit to ours. [Chaudhari et al. 2019] introduced a variant of SGD guided by local entropy to bias the dynamics into wider minima. [Izmailov et al. 2018] demonstrated how a simple algorithm of stochastically averaging samples from the limiting dynamics of a network can improve generalization performance. [Fort et al. 2019] demonstrated how ensembling trained models can result in improved test performance. While algorithm development is not the focus of our work, we believe that our careful and precise understanding of SGD in simple settings and limiting dynamics of deep neural networks will similarly provide insight for future work in this direction.  It is important to emphasize that in this work, we take a phenomenological route that has proven to be promising to unlocking mysteries in deep learning (for example Double Descent (Belkin et al., 2018) or Spherical Motion Dynamics (Wan et al., 2020)).  By identifying phenomena, explaining them in a simpler setting, and providing evidence of these predictions at scale we are furthering our understanding of SGD and deep learning.  In particular, understanding the role of stochasticity in deep learning, even in the restricted setting of the limiting dynamics, is an area of quite active research right now and we see our work as an important addition to this literature.
>
> (2) **The novelty and originality of our results:**
> We understand that our theoretical work directly builds upon a series of related works, but we were very careful to explicitly state what our contributions are and how they relate to these previous works in the sections titled “significance” and “originality” on page 9 in addition to the related work section. Based on *our* stated contributions and originality, could you elaborate what you mean when you say “the results derived from the Ornstein-Uhlenbeck process do not go beyond the previous work”?  What results and previous works are you referencing?
>
> (3) **Unclear presentation:**
> We agree with you that our stylistic choice of using qualifier such as "surprising", "contrary to common intuition", "nonintuitive" to highlight observations in the text can be mistaken as claims of novelty and thus we will use these terms more sparingly.  That said (style aside), we were very careful to clarify previous results from our work, which is why we devoted over a page to the related work and added the sections “significance” and “originality” on page 9.  Do you have other suggestions that you think might improve the clarity in this regard?  As for clearly separating “theoretical predictions from numerical measurements (Fig. 6)”, we think you might be misunderstanding aspects of section 8.  For the local displacement we derive a theoretical expression (equation 11) using a single experiment estimate a scalar quantity and then predict over a large sweep of different hyperparameters (the black dotted line on the top row of fig. 6).  We then train 60 independent experiments with unique hyperparameter combinations and plot their empirical values as the colored circles in the figure.  As for the exponent of anomalous diffusion, we did not claim to predict the exponent directly, which is why we write “unfortunately, it is not immediately clear how to extract the explicit exponent $c$ from equation (12)”, rather our goal was to “predict overall trends in the diffusion exponent $c$.”  We changed the title of this section from “Predicting the diffusive behaviour...” to “Understanding the diffusive behaviour…”  We hope this will help avoid confusion.
>
> We thank you again for your review and detailed comments and questions.  Given our responses to your three concerns we hope you would consider raising your score to accept.  If you do not agree we hope to have a discussion so we can understand your reservations and hopefully clarify our work.  Thank you!

---

> > ### Comment · Reviewer_4wwS · 2021-11-11
> > **Thank you for your reply**
> >
> > Thank you for responding to my comments.
> >
> > # Regarding the references you mention in the reply
> >
> > As I wrote in my initial review, "I welcome the direction of the present study, trying to understand the dynamics of deep neural networks via simpler models." However, I am not sure what to make with this list of references, most of which seem at best tangentially related to the present paper.  [Krizhevsky, 2014] does not analyse an SDE, he analyses SGD updates directly, and that's not the main point of the paper anyway. I assume by [Goyal '17] you refer to the Training Imagenet in 1hr paper - again, this paper is focusing on quick training, as the title suggests. They look directly at SGD updates, but do not analyse SDE approximations - so how are these papers relevant for this discussion? I also don't see at all the connection to the double descent [Belkin '18], which is a phenomenon about generalisation, completely unrelated to limiting dynamics, or to the entropy-SGD algorithm of Chaudhari et al., which again aims to improve generalisation by selecting wider minima *during training*, not once training has converged. What is the connection of the present paper to ensembling [Fort '19] ?! Etc. There are many papers studying simplified models, but I don't see how this is an argument for (or indeed against) accepting this paper.
> >
> > # Novelty / clarity of the presentation / etc:
> >
> > I appreciate your using qualifiers like "surprising" more sparingly (though I don't see any chances in the manuscript?) "Style aside", as you write, I think the issues here run deeper and are indeed not stylistic. As I tried to explain in my review, you open the paper with an observation the you claim to be novel, but it is not, cf. the references I gave above. Several other referees pointed this out as well. I don't think this is fixed by discussing these papers in a "Significance" section on page 9, this needs to be clarified from the beginning. I have similar reservations about the discussion on modified losses, which permeate the literature - I gave some references above, some of which you also cite; I learnt about Ref arXiv:1803.01927 in another review, and I am sure that there are many others. Again, previous work, especially if it is so closely related, shouldn't be relegated to page 9 - it should form the starting point of the article, and be discussed concurrently.
> >
> > > Based on our stated contributions and originality, could you elaborate what you mean when you say “the results derived from the Ornstein-Uhlenbeck process do not go beyond the previous work”? What results and previous works are you referencing?
> >
> > I think that most of the things you discuss in your theoretical appendices D to G concerning the analysis of the OU are standard, and can be found in various books: Gardiner Sec 5.3 (cited in my review), or Risken's "Fokker-Planck equation". I am thinking in particular of the calculation of the mean and variance of the OU process, Eq. 16 an 18, discussion of the Lyapunov equation, the decomposition of the drift matrix is discussed in Kwon et al. as you mention. The variational formulation, e.g. Eq. 33 is usually found as "Potential conditions", but is also found in deep learning papers (Chaudhari et al.) etc. Appendix G calculates the mean and variance of a process under the assumption that its stationary distribution is Gaussian - again, these are known calculations.
> >
> > # Experimental section
> >
> > I understood Sec. 8 exactly in the way that you summarise it here. I think that changing the title of the section is a step in the right direction.
> >
> > # Summary
> >
> > I thank you again for your responses, however my original concerns have not been alleviated to the point where I would feel comfortable raising my score.

---

> > > ### Author Response · Authors · 2021-11-12
> > > **Thank you for your reply**
> > >
> > > Thank you for your fast reply.
> > >
> > > **Refrences:** Your first concern was “the implications of the studied dynamics on learning or representations in neural network remain unclear to me”.  We provided these references as examples of algorithmic/empirical works made in deep learning based on analysis of SGD similar in style to ours. We understand now that your concern is more about our focus on the limiting dynamics, not the style of our analysis.  However, consider [Izmailov et al., 2018], one of the papers we mentioned in our list.  The authors of this work found that “that simple averaging of multiple points along the trajectory of SGD, with a cyclical or constant learning rate, leads to better generalization than conventional training”.  This work studies pretrained models (in fact the exact same default Pytorch models we studied) and finds “SWA [Stochastic Weight Averaging] provides consistent improvement by 0.6-0.9% over the pretrained models.”  Follow up works on SWA, also leveraging the limiting dynamics of pretrained neural networks, include [Maddox et al. 2019] and [Izmailov et al., 2020].  Prior works to SWA, also studying pretrained neural networks, include [Garipov et al. 2018], who found that Fast Geometric Ensembling (FGE) of a pretrained neural network could improve the final performance (this is the connection to ensembling).  As explained by Garipov et al., “We used a pretrained model with top-1 test error of 23.87 to initialize the FGE procedure. We then ran FGE for 5 epochs....the top-1 test error-rate of the final ensemble was 23.31. Thus, in just 5 epochs we could improve the accuracy of the model by 0.56 using FGE”
> > > All these works demonstrate that even “after” training state-of-the-art neural networks, there is still lots of room for performance improvement.  Thus, the theoretical and empirical study of limiting dynamics is not trivial and has yielded useful results.
> > >
> > >  **Novelty:** You’re second concern was “the results derived from the Ornstein-Uhlenbeck process do not go beyond the previous work to an extent that would warrant the publication at ICLR in my opinion” and you clarified in your response that specifically, you “think that most of the things you discuss in your theoretical appendices D to G concerning the analysis of the OU are standard, and can be found in various book”. So the derivations and background material in *the appendix* is what concerns you because similar derivations can be found in various textbooks and other references?  Would you rather us remove these sections from the appendix and simply reference the reader to these sources?  It is quite surprising that derivations in the appendix would be concerning.
> > >
> > > **Clarity:**
> > > Your third concern is “ the unclear presentation of the results which doesn't always clearly separate previous work from the presented results, or theoretical predictions from numerical measurements (Fig. 6)”.
> > > We are glad that removing certain adjectives and adverbs will make the paper more readable and clear.  We will update the current PDF shortly once we hear back from other  reviewers.  You write “previous work, especially if it is so closely related, shouldn't be relegated to page 9 - it should form the starting point of the article, and be discussed concurrently”. This doesn’t seem to be an accurate characterization of our work at all.  We discuss a broad array of related work for over a page in the very beginning of our work, and cite continuously throughout sections where it seems relevant in addition to the section on page 9.  Regardless, we will add more inline references.
> > > As far as we were aware, the observations of constant local and global displacement we presented in section 2 were novel, despite similar observations made in previous literature, which we cite in this section.  We will be adding additional references to [arXiv:1803.06969] now that we are aware of this work.  While we were not aware of these observations previously, the goal of this section was to motivate the question *where is the network moving to and why?*, the central question of our paper.  And indeed the explanation we come to and empirically test in section 8 is novel.
> > > Given that you understand our experimental section then are you still concerned that our results don't “clearly separate...theoretical predictions from numerical measurements (Fig. 6)”?  If you are still concerned, could you explain what specifically concerns you?
> > >
> > > Thank you again for your review and reply.  We hope this alleviates your concerns and you will consider raising your score to accept.

---

> > > > ### Comment · Reviewer_4wwS · 2021-11-14
> > > > **Thank you for your further explanations.**
> > > >
> > > > **References** I'm neither commenting on the entire research direction, or the approach in general, although I'd like to repeat that I think that "trying to understand the dynamics of deep neural networks via simpler models" is a valuable approach in general. Here, I am reviewing only the given paper, hence I still don't know what to do with all these references that are not related to the paper. My original concern remains that I struggle to see implications for either training error or test error, or the representations learnt by the network. Hence I'm afraid the paper is of limited interest to the ICLR community.
> > > >
> > > > **Novelty** Key technical results in this paper, following the abstract, seem to be "the derivation of exact, analytical expressions for the phase dynamics" and that you show "using the Fokker-Planck equation [...] that [...] the key ingredient driving these dynamics is not the original training loss" Hence my point is not whether or not you put derivations in an appendix or not; it's that these manipulations are standard results regarding OU processes that by themselves they do not warrant publication at ICLR in my eyes, especially since several of the ensuing observations, such as the existence of a modified losses for SGD, have been identified in very similar setups in the literature, as has been discussed above and by several other reviews at length already.
> > > >
> > > > As regards the novelty of the results and the clarity of the presentation, I feel like the issue has been discussed in detail in several of the reviews including mine, so I don't want to reiterate the points here another time. I'd like to thank the authors for the exchange, which has clarified a few points.

---

### Official Review · Reviewer_erMi · 2021-11-02

**Correctness:** 3
**Technical Novelty And Significance:** 3
**Empirical Novelty And Significance:** 3
**Recommendation:** 6
**Confidence:** 3

**Main Review:**

**Strength**

1. The theoretical results are solid and well-motivated;
2. The experiments are thorough, and consistent with theoretical results;
3. The contribution of this work are carefully clarified.

**Weakness**

The only pity is that all the theoretical results are derived from a linear regression model. The impact of this work will significantly increase if these interesting theoretical results can be derived in more general cases.

**Further concerns**

I notice some disruptive claims are raised in the last paragraph. Please can you elaborate them in order to raise my score? The following two concern me the most:
- “the network stays within a local region” is wrong;
-  "Finally, the intuition that faster learning rates would lead to faster global displacement in parameter space is also wrong; instead induced velocity anti-correlations lead to slower global displacement"

**Summary Of The Paper:**

This paper derive a continuous model for SGD in linear regression settings to build understanding on some interesting phenomenon of deep neural network trained with SGD.

**Summary Of The Review:**

In this paper, the insight from the theoretical results can match the empirical observations pretty well. Their results certainly deepens our understanding on the learning dynamics of modern neural network.

---

> ### Author Response · Authors · 2021-11-10
> **Thank you for your review**
>
> Thank you for your review and we are glad to hear that our “results certainly deepens our understanding on the learning dynamics of modern neural network”.
>
> Here we elaborate on the two claims that concerned you:
>  - **The network stays within a local region is wrong:** The intuition that the model would converge in parameter space is wrong as we demonstrate empirically in figure 1, all trained models monotonically move away from their initial trained state.  Maybe you found the wording “within a local region” confusing or too vague?  We agree and will remove this part of the sentence such that it now reads “Natural intuitions, such as “the network converges to a local region in parameter space” is wrong; networks anomalously diffuse away.”
>  - **Faster learning rates would lead to faster global displacement in parameter space is also wrong:** This claim is explained in section 8 and shown in figure 6 empirically where you can see that for larger learning rates, while the expectation of the local displacement increases, the rate of anomalous diffusion actually decreases.  As explained in section 8 under *Exponent of anomalous diffusion* (p. 8), this is caused by “a larger learning rate leads to underdamped oscillations, and the resultant temporal velocities’ anticorrelations reduce the exponent of anomalous diffusion”.  In other words, a  large learning rate will lead to lots of oscillations which will actually slows down global movement.
>
> As for your remark that “the only pity is that all the theoretical results are derived from a linear regression model”, we agree that it would be ideal if we could derive similar theoretical results in a more complex setting.  Of course, generally speaking in deep learning, theory can only be developed in simpler settings and applied or tested empirically in the more complex settings.  This is what we did in this work and made sure to clearly state all assumptions and the assumptions needed to make our theory apply.
>
> We hope this helps and that you will consider raising your score.  We are happy to clarify any other questions you might have and hope to have a conversation with you over the next week.

---

### Author Response · Authors · 2021-11-22
**Thank you to all Reviewers**

First we would like to thank all reviewers for their careful and detailed comments.  We greatly appreciate the time and effort you spent reviewing our paper, which we are certain will make our work better. We have updated the manuscript reflecting the changes we discussed through the review process.  In particular, you will find:

 - We removed the word “Rethinking” from the title
 - We removed the claim of observing anomalous diffusion first, as evidenced in the second line of the abstract, “As observed previously…”
 - We added new citations including [Baity-Jesi et al. 2018], [Chen et al., 2021], [Rotskoff, Vanden-Eijnden 2018], [Chizat, Bach 2018], [Mignacco et al. 2020], [Mannelli et al. 2020], [Mignacco et al. 2021], [Mannelli, Urbani 2021], [Ghorbani, Krishnan, Xiao 2019].
 - We updated figures as suggested (Fig. 1 x axis has more ticks, Fig. 6 labeling is clarified)
 - We changed section 8 header from “Predicting diffusive behavior...” to “Understanding the diffusive behavior...”
 - We removed the “this is wrong” claims from the conclusion.
 - We removed adjectives and adverbs that embellish the paper and complicate the claims of novelty.

Thank you again for the time you spent reviewing this work and we hope that despite the raised issues we tried to address in our revision, you find the value of this paper worth accepting.

---

### Decision · Program_Chairs · 2022-01-20

**Decision:**

Reject

**Comment:**

The authors study the limiting dynamics of a simple linear regression model. They use an underdamped Langevin equation which is quite common in the literature. Although the reviewers welcome the direction and the attempt to understand the dynamics of a simple model, the novelty of the paper is limited. As an example, the paper shows that the key ingredient driving these dynamics is not the original training loss, but a modified loss. As pointed out by the reviewers, this has already been pointed out in multiple papers. One important problem is the tendency of the paper to oversell the results (including the title), which makes it difficult to clearly separate the contributions made in the paper. After a discussion with the reviewers, the overall feeling did not change.

I can therefore not recommend acceptance. I strongly recommend the authors do a significant rewrite of the paper in order to clearly separate what contributions are truly novel and also improve the discussion of prior work.